# Reducing Excessive Margin to Achieve a Better Accuracy vs. Robustness Trade-off

**Rahul Rade**
ETH Zürich, Switzerland
`rarade@ethz.ch`

**Seyed-Mohsen Moosavi-Dezfooli**
Imperial College London, UK
`seyed.moosavi@imperial.ac.uk`

## Abstract

While adversarial training has become the de facto approach for training robust classifiers, it leads to a drop in accuracy. This has led to prior works postulating that accuracy is inherently at odds with robustness. Yet, the phenomenon remains inexplicable. In this paper, we closely examine the changes induced in the decision boundary of a deep network during adversarial training. We find that adversarial training leads to *unwarranted* increase in the margin along certain adversarial directions, thereby hurting accuracy. Motivated by this observation, we present a novel algorithm, called *Helper-based Adversarial Training (HAT)*, to reduce this effect by incorporating additional *wrongly* labelled examples during training. Our proposed method provides a notable improvement in accuracy without compromising robustness. It achieves a better trade-off between accuracy and robustness in comparison to existing defenses. Code is available at `https://github.com/imrahulr/hat`.

## 1 Introduction

Despite the remarkable success of modern deep networks on computer vision tasks, they are notoriously fragile to small, imperceptible changes in the input (Szegedy et al., 2014). Ever since the introduction of $\ell_p$-norm based adversaries (Goodfellow et al., 2015; Moosavi-Dezfooli et al., 2016), a growing body of researchers has focused on developing solutions for improving the robustness of DNNs. At present, adversarial training (Madry et al., 2018) remains the most effective approach for robustification. Nevertheless, it is known to cause an undesirable reduction in accuracy, thus leading to the much-debated trade-off between accuracy and robustness (Tsipras et al., 2019; Yang et al., 2020). Subsequently, in an endeavour to alleviate the accuracy-robustness trade-off, several refinements have been proposed to the vanilla adversarial training, e.g., Zhang et al. (2019); Wang et al. (2020); Rice et al. (2020); Wu et al. (2020); Zhang et al. (2021). However, as shown in Fig. 1, the progress has been rather incremental over the last few years.

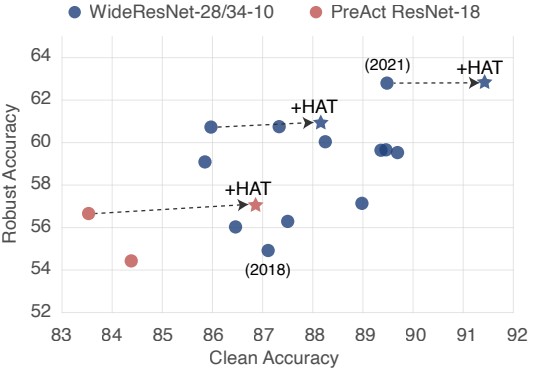

Figure 1: Accuracy-robustness trade-off obtained by prominent defenses on CIFAR-10 against $\ell_\infty$ perturbations. We compile top models which use additional data and are available in Croce et al. (2020). From 2018 to 2021, the accuracy for robust WideResNet-28/34-10 models has only increased by around $2.5\%$. While our proposed HAT, combined with recent approaches, advances clean accuracy further by about $2\text{-}4\%$.

Notwithstanding these recent advances, closing the large gap between accuracy and robustness still remains an open challenge and every modest improvement in this regard remains significant.

In this paper, we take a step towards demystifying and improving the aforementioned trade-off. To this end, we take a closer look at the effect of training with adversarial examples on the geometry of decision boundaries learnt by deep networks. Specifically, relying on the perspective proposed by Ortiz-Jimenez et al. (2020) which relates the directional margin with the discriminative features

used by a network, we uncover an unwanted consequence of adversarial training. That is, we identify that adversarial training leads to a superfluous increase in the margin along the adversarial directions of the input space computed for a regularly trained network. We refer to these directions as *initial*

adversarial directions[1]. In other words, the network becomes excessively robust (compared to the attack radius $\varepsilon$) along the directions which are otherwise pivotal for regular networks, and hence the accuracy drops. Further, we establish an empirical connection between the margin along initial adversarial directions and the classification performance of the network. In particular, a smaller margin along these directions corresponds to a higher accuracy. Thus, while the excessive margin is not essential for robustness, it severely derails the performance on clean samples. This naturally raises the following question:

*Can we foresee an improvement in accuracy by reducing the margin along initial adversarial directions while preserving the same level of robustness?*

To address this question, we propose a heuristic adversarial training scheme to reduce the directional margin and thus, achieve a better accuracy without sacrificing robustness. We call this algorithm, *Helper-based Adversarial Training (HAT)*, the name derived from the fact that we incorporate additional training examples comprising of overly perturbed adversarial images (possibly) wrongly labelled using a standard trained network to *help* impede excessive directional robustness and hence, improve clean accuracy (see Fig. 2). We also provide an extensive analysis of HAT and compare it with state-of-the-art AT methods. The main contributions of this paper include:

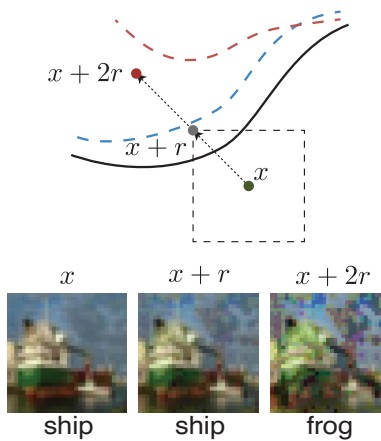

Figure 2: (Top): Illustration depicting the purpose behind introducing helper examples labelled by a standard classifier. Solid black line: standard network. Dashed red line: adversarially trained network. Dashed blue line: desired decision boundary. (Bottom): $x + 2r$ is visually dissimilar from $x$ and the classifier is not needed to be robust to such large distortions.

- We demonstrate that adversarial training leads to a large margin along initial adversarial directions. In addition, we identify a direct connection between the excessive margin and the reduction in clean accuracy caused by adversarial training.

- We propose *Helper-based Adversarial Training (HAT)* to explicitly reduce the excessive margin by augmenting the training data with properly-labelled helper examples.

- Experimentally, we show that HAT consistently improves accuracy without sacrificing robustness. Notably, in combination with recent improvements from Rebuffi et al. (2021), HAT sets a new state-of-the-art and improves clean accuracy by 3.33% on CIFAR-10 and 4.63% on CIFAR-100 using PreAct ResNet-18 against $\ell_\infty$ perturbations of size $8/255$.

## 1.1 RELATED WORK

**Improving adversarial robustness.** To mitigate the adversarial susceptibility of DNNs, a myriad of defense methodologies (Papernot et al., 2016; Tramèr et al., 2018; Kannan et al., 2018; Wang et al., 2019; Xiao et al., 2020) have been proposed. Yet, many of them have been shown to provide superficial robustness by introducing gradient obfuscation (Athalye et al., 2018), hence calling for a cautious evaluation of robustness (Tramer et al., 2020; Croce et al., 2020). Nonetheless, adversarial training (AT) (Madry et al., 2018) or training with worst-case inputs has been the most popular approach for improving the robustness of DNNs albeit at the cost of a sizeable drop in accuracy. Notably, AT has been able to withstand newer sophisticated adversaries (Gowal et al., 2019; Croce & Hein, 2020). Another prominent defense is TRADES (Zhang et al., 2019), which provides a systematic way to control the trade-off between accuracy and robustness by solving:

$$\arg\min_{\boldsymbol{\theta}} \sum_{i=1}^{n} \text{CE}(y_i, f_{\boldsymbol{\theta}}(\boldsymbol{x}_i)) + \beta \max_{\boldsymbol{\delta}_i:||\boldsymbol{\delta}_i||_p \leq \varepsilon} \text{KL}(f_{\boldsymbol{\theta}}(\boldsymbol{x}_i), f_{\boldsymbol{\theta}}(\boldsymbol{x}_i + \boldsymbol{\delta}_i)), \tag{1}$$

[1]The reason for such terminology will become clear in the later sections.

where CE is the cross-entropy loss, KL is the Kullback-Leibler divergence and $\beta$ controls the trade-off between accuracy and robustness. More recent attempts to improve the accuracy vs. robustness trade-off have predominantly dealt with refining adversarial training by (i) modifying the objective function (Wang et al., 2020; Zhang et al., 2020; 2021), (ii) early-stopping (Rice et al., 2020), (iii) heuristics and hyperparameter selection (Pang et al., 2021; Gowal et al., 2021), (iv) extra data (Hendrycks et al., 2019; Alayrac et al., 2019; Carmon et al., 2019; Rebuffi et al., 2021; Sehwag et al., 2021), and (v) weight initialization for training with large perturbations (Shaeiri et al., 2020). Despite these advancements, the problem is far from being solved with a huge gap between accuracy and robustness on practical image recognition benchmarks. We revisit this issue in our paper and build upon TRADES to further ameliorate the trade-off.

**Understanding adversarial vulnerability.** A concurrent stream of prior works has provided plausible reasons for the existence of adversarial examples. Adversarial vulnerability of DNNs has been accredited to the existence of highly generalizable, but non-robust features in the data which are exploited by DNNs to achieve state-of-the-art performance (Jetley et al., 2018; Ilyas et al., 2019). Besides, Ortiz-Jimenez et al. (2020) devised a novel experimental framework which relates the distance of data points to the decision boundary of a DNN with the discriminative features used by the DNN in order to understand the mechanism behind AT.

## 2 PRELIMINARIES

Consider the input space $\mathcal{X} \subseteq \mathbb{R}^d$. Let $f_{\boldsymbol{\theta}} : \mathcal{X} \rightarrow \mathbb{R}^C$ represent a deep neural network classifier parameterized by $\boldsymbol{\theta}$, where $C$ is the number of output classes. Let $F_{\boldsymbol{\theta}}(\boldsymbol{x}) = \arg\max_k f_{\boldsymbol{\theta}}(\boldsymbol{x})_k$ be the class label predicted by $f_{\boldsymbol{\theta}}$ for any $\boldsymbol{x} \in \mathcal{X}$, where $f_{\boldsymbol{\theta}}(\boldsymbol{x})_k$ is the $k^{\text{th}}$ component of $f_{\boldsymbol{\theta}}(\boldsymbol{x})$.

**Margin.** Given a classifier $F_{\boldsymbol{\theta}}$, input $\boldsymbol{x}$ and an unit vector $\hat{\boldsymbol{r}} \in \mathbb{S}^{d-1}$, we define margin $\mu$ at $\boldsymbol{x}$ along the direction $\hat{\boldsymbol{r}}$ as:

$$\mu(\boldsymbol{x}, \hat{\boldsymbol{r}}) = \arg\min_{\alpha} |\alpha| \ \ \text{s.t.} \ \ F_{\boldsymbol{\theta}}(\boldsymbol{x} + \alpha\hat{\boldsymbol{r}}) \neq F_{\boldsymbol{\theta}}(\boldsymbol{x}) \tag{2}$$

Additionally, note that we refer to a deep network trained only on clean samples as *standard network* and a network trained via adversarial training as *robust network*. Besides, we reuse the definitions of clean (natural) and robust (adversarial) accuracy as stated by Zhang et al. (2019).

**Initial adversarial directions.** Given a standard network $f_{\boldsymbol{\theta}}$, input dataset $\{(\boldsymbol{x}_i, y_i)\}_{i=1}^n$, we define the set of initial adversarial directions as $\mathcal{R}_{\text{init}} = \{\boldsymbol{r}_i/||\boldsymbol{r}_i||_2\}_{i=1}^n$ where $\boldsymbol{r}_i$ is obtained by solving:

$$\boldsymbol{r}_i = \max_{\boldsymbol{\delta}_i : ||\boldsymbol{\delta}_i||_p \leq \varepsilon} \ell(y_i, f_{\boldsymbol{\theta}}(\boldsymbol{x}_i + \boldsymbol{\delta}_i)). \tag{3}$$

where, $\ell(\cdot, \cdot)$ is an arbitrary loss function e.g. cross-entropy (CE). This optimization is usually solved via a multi-step procedure called projected gradient descent (PGD) (Madry et al., 2018).

## 3 ADVERSARIAL TRAINING INTRODUCES EXCESSIVE INVARIANCE

We begin our analysis by examining the effect of adversarial training (AT) on the decision boundary of DNNs. Via novel analysis on a synthetic dataset and CIFAR-10 (Krizhevsky, 2009), we show that AT triggers a superfluous increase in the margin along the initial adversarial directions as compared to the nominal increase required to attain robustness. In addition, we provide empirical evidence which signifies a connection between the increase in margin and reduction in clean accuracy.

**Toy problem.** First, we study a toy setting to shed some light on the phenomenon of excessive directional margin caused by AT. We construct a 3-d binary classification dataset drawn from two distributions which live on two noisy concentric circles of different radii in the $x_1$-$x_2$ plane and being linearly separable along the third dimension $x_3$. In particular, $x_1 = \rho_i \cos(z) + \epsilon_1$, $x_2 = \rho_i \sin(z) + \epsilon_2$ and $x_3 \sim \mathcal{U}(\alpha_i, \beta_i)$ where $z \sim \mathcal{U}(0, 2\pi)$ and $\epsilon_1, \epsilon_2 \sim \mathcal{N}(0, \sigma^2)$ where $i = 1, 2$ for class 1 and 2 respectively. We train a single hidden-layer MLP via both standard training and adversarial training. Fig. 3 visualizes the decision regions with both the training procedures. It is

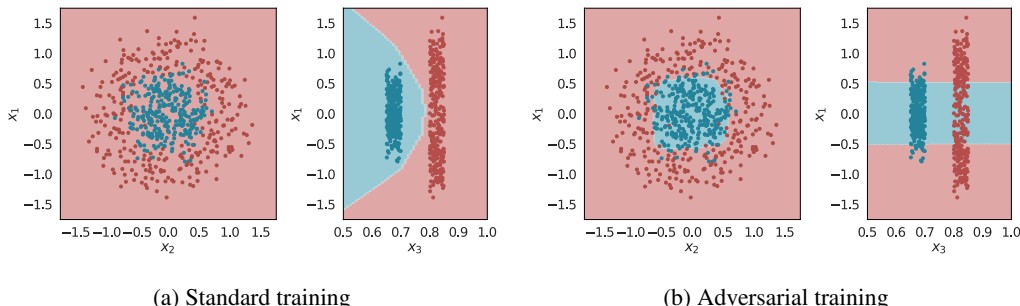

(a) Standard training                (b) Adversarial training

Figure 3: Decision boundary learnt by MLP visualized in two dimensions: $x_1$-$x_2$ and $x_1$-$x_3$ respectively. Adversarial training improves robustness substantially from $41\%$ to $74\%$, yet causes about $9\%$ drop in accuracy. For each left subplot, $x_3 = 0.85$ and for each subplot on the right, $x_2 = 0.4$.

evident that the network primarily uses $x_3$ to achieve zero classification error when trained using standard training, but the resulting model performs poorly in terms of robustness. In contrast, when we use AT, the learned decision boundary is completely different from that in the standard case. Here, the network becomes reasonably invariant along $x_3$ (Fig. 3b), thus causing the directional margin along $x_3$ to tend to $\infty$. This enables the network to attain robustness at the cost of a small increase in classification error.

**Evidence on CIFAR-10.** Next, we illustrate that a similar phenomenon occurs in the case of state-of-the-art deep networks trained on CIFAR-10. To this end, we measure directional margin (as defined by Eq. 2) to the classification boundary which allows us to partly capture the geometry of decision boundary. We restrict ourselves to the following setting. We take a ResNet-18 (He et al., 2016) trained until convergence on CIFAR-10 (achieving $94.6\%$ accuracy and $0\%$ robustness to $\ell_\infty$-PGD perturbations on the test set) and then fine-tune it using AT with adversarial examples. This framework allows us to study the evolution of the decision boundary caused by AT in comparison to that learnt by a standard network. We use $\ell_\infty$-PGD with norm $\varepsilon = 8/255$ for training. The network attains $83.3\%$ accuracy and $51.6\%$ robustness on the test set after adversarial fine-tuning. Please refer to App. B.1 for more details of the experimental setup.

Meanwhile, during adversarial fine-tuning, we track the margins along the adversarial directions found by PGD to shed some light on the learning dynamics. Suppose $\mathcal{R}_k = \{\boldsymbol{r}_i^k / \|\boldsymbol{r}_i^k\|_2\}$, where $\boldsymbol{r}_i^k$ denotes adversarial perturbation found by PGD at $k^{\text{th}}$ epoch of adversarial fine-tuning for the input sample $\boldsymbol{x}_i$. Thus, $\mathcal{R}_{\text{init}} = \mathcal{R}_0$ represents the set of initial adversarial directions. We hypothesize that during adversarial training, the network becomes excessively robust to these initial adversarial directions $\mathcal{R}_0$ while slightly shifting the decision boundaries along other adversarial directions $\mathcal{R}_{k\geq 1}$. Fig. 4 illustrates the margins along $\mathcal{R}_0$, $\mathcal{R}_1$ and $\mathcal{R}_5$ before and after adversarial fine-tuning respectively. The dashed red line indicates the value of $\varepsilon$ used for training, i.e., any allowable pertur-

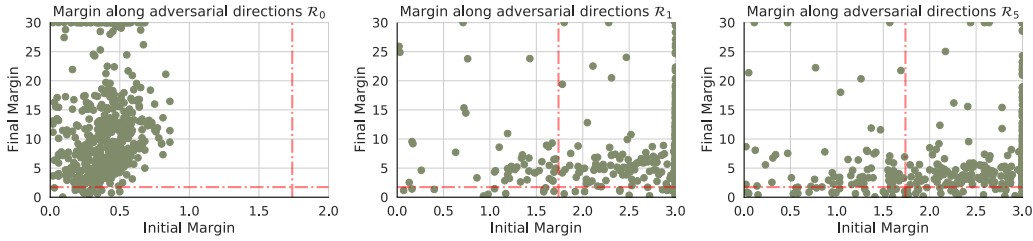

Figure 4: Final margins for adversarially trained model vs. initial margins before the start of adversarial fine-tuning on CIFAR-10 along $\mathcal{R}_0$ ($= \mathcal{R}_{\text{init}}$), $\mathcal{R}_1$ and $\mathcal{R}_5$ respectively. Dashed line indicates the value of $\varepsilon$ used during training and evaluation. The increase in margin along $\mathcal{R}_0$ is much larger than that along other directions $\mathcal{R}_1$ and $\mathcal{R}_5$.

bation $\boldsymbol{r}$ has $||\boldsymbol{r}||_\infty \leq 8/255$ or equivalently $||\boldsymbol{r}||_2 \leq 1.74$. Intuitively, one might expect adversarial fine-tuning to cause small shifts in the decision boundary so that the margin becomes greater than 1.74, and attain robustness. However, this is not the case in practice. Intriguingly, the classifier instead resorts to becoming largely insensitive along $\mathcal{R}_0$ for some data points as evident in Fig. 4 while undergoing small shifts along other directions $\mathcal{R}_{k \geq 1}$. We also observe a decrease of $11.3\%$ in clean accuracy after fine-tuning.

**Connection between margin along $\mathcal{R}_{\textbf{init}}$ and clean accuracy.** We now provide a two-fold argument which justifies the following hypothesis: "The drastic rise in the margin along $\mathcal{R}_{\text{init}}$ is directly correlated to the observed reduction in accuracy". In fact, a larger margin contributes to a larger drop in accuracy. (i) Firstly, we complement our hypothesis with the following observation by Ortiz-Jimenez et al. (2020). The directions of input space with small margins and in turn, the initial adversarial directions in the case of standard network, are associated with discriminative features learnt by the network. We believe that these directions are crucial for the performance of the network. Thus, a drastic directional margin along these

Table 1: Median margin along $\mathcal{R}_{\text{init}}$ and the corresponding clean and robust accuracy with TRADES on CIFAR-10 test set for different values of $\beta$. Robust accuracy is evaluated with AutoAttack (Croce & Hein, 2020).

| $\beta$ | Median Margin | Clean | Robust |
|-----|-----|-----|-----|
| 1.0 | 8.3 | 88.1 | 43.8 |
| 2.0 | 9.3 | 85.6 | 46.3 |
| 3.0 | 9.7 | 84.7 | 47.9 |
| 4.0 | 10.3 | 83.6 | 48.5 |
| 5.0 | 10.5 | 82.9 | 48.8 |

directions contributes to the drop in classification accuracy. (ii) We train a robust network on CIFAR-10 using TRADES with different values for the trade-off parameter $\beta$. (Please refer to Sec. 5.1 for the exact experimental setup.) As $\beta$ increases, we observe an increase in the margin along $\mathcal{R}_{\text{init}}$ (computed on a subset of 1024 examples from the CIFAR-10 test set) and a corresponding reduction in clean accuracy (see Table 1). This further corroborates our hypothesis.

## 4 HELPER-BASED ADVERSARIAL TRAINING

As demonstrated in Sec. 3, AT triggers an unwarranted increase in the margin along initial adversarial directions, thus hindering the network from using highly discriminative features in those directions. Note that a large margin is not a sufficient condition for $\ell_p$-based $\varepsilon$-robustness. A trivial example of this is a constant classifier which has an infinite margin, but has both poor robustness and poor accuracy. Instead, the excessive rise in margin translates to large changes in the geometry of decision boundary compared to a standard network and causes a corresponding drop in clean accuracy. We here base our algorithm on the intuition that slightly pushing the decision boundary away from data should suffice for attaining robustness as opposed to the large shifts observed. Our results in Sec. 5 suggest that HAT, by reducing margin, can indeed lead to improved performance.

Now, we introduce our proposed algorithm, *helper-based adversarial training (HAT)*, to reduce the excessive directional margin observed in practice. We aim to preserve certain geometric properties of a standard trained network viz. the predictive power along adversarial directions while learning a robust model. To this end, we add additional training examples, called helper examples, which are generated on-the-fly during the adversarial training procedure. In particular, a helper example is constructed by extrapolating the adversarial perturbation found during training and is (possibly wrongly) labelled by a standard network (see Fig. 2). Formally, we define a helper example as:

**Helper examples.** Given an input sample $(\boldsymbol{x}_i, y_i)$, a standard network $f_{\boldsymbol{\theta}_{\text{std}}}$, a robust network iterate $f_{\boldsymbol{\theta}_{\text{rob}}^k}$ at $k^{\text{th}}$ training iteration and adversarial example $\boldsymbol{x}_i'$ computed by the adversary $\varphi$ for $f_{\boldsymbol{\theta}_{\text{rob}}^k}$, the corresponding helper example is given by $(\tilde{\boldsymbol{x}}_i, \tilde{y}_i)$ where

$$\tilde{\boldsymbol{x}}_i = \boldsymbol{x}_i + 2\boldsymbol{r}_i, \ \boldsymbol{r}_i = \boldsymbol{x}_i' - \boldsymbol{x}_i \text{ and } \tilde{y}_i = \arg\max_k f_{\boldsymbol{\theta}_{\text{std}}}(\boldsymbol{x}_i')_k$$

The motivation behind this definition is illustrated in Fig. 2 where it is evident that we can stimulate a slight push to the decision boundary along any adversarial direction by making the network predict the correct label $y_i$ at adversarial example $\boldsymbol{x}_i'$ and have it predict $\tilde{y}_i$ (often $\tilde{y}_i \neq y_i$) at helper example $\tilde{\boldsymbol{x}}_i$ to preserve the discriminative characteristics as modelled by a standard network. Defining the helper examples as $\boldsymbol{x} + 2\boldsymbol{r}_i$ is a heuristic choice that provides a good compromise between the helper example being sufficiently dissimilar (from $\boldsymbol{x}$) for the model to assign a different label (see Fig. 2 for a visual example), and not too dissimilar to affect the performance on other clean samples. In

---

**Algorithm 1** Helper-based Adversarial Training

**Input:** Training dataset $\mathcal{D} = \{(\boldsymbol{x}_i, y_i)\}_{i=1}^n$
**Parameter:** Batch size $m$, learning rate $\eta$, weight of robust loss $\beta$, weight of helper loss $\gamma$, attack radius $\varepsilon$, attack step size $\alpha$ and number of attack iterations $K$

1: Train a network $f_{\boldsymbol{\theta}_{\text{std}}}$ via standard training on $\mathcal{D}$ i.e., $\boldsymbol{\theta}_{\text{std}} \leftarrow \arg\min_{\boldsymbol{\theta}} \sum_{i=1}^n \text{CE}(y_i, f_{\boldsymbol{\theta}}(\boldsymbol{x}_i))$
2: Randomly initialize the network parameters $\boldsymbol{\theta}_{\text{HAT}}$
3: **repeat**                                                                        ▷ Train a robust classifier
4:     Sample a mini-batch $\{(\boldsymbol{x}_{i_j}, y_{i_j})\}_{j=1}^m$ from $\mathcal{D}$
5:     **for** $j = 1, 2, \ldots, m$ **do**
6:         $\boldsymbol{x}'_{i_j} \leftarrow \boldsymbol{x}_{i_j} + 0.001 \cdot \mathcal{N}(0, I)$                          ▷ Construct adversarial example
7:         **for** $k = 1, 2, ..., K$ **do**
8:             $\boldsymbol{x}'_{i_j} \leftarrow \prod_{\mathcal{B}(\boldsymbol{x}_{i_j}, \varepsilon)} (\boldsymbol{x}'_{i_j} + \alpha \cdot \text{sign}(\nabla_{\boldsymbol{x}'_{i_j}} \text{KL}(f_{\boldsymbol{\theta}_{\text{HAT}}}(\boldsymbol{x}_{i_j}), f_{\boldsymbol{\theta}_{\text{HAT}}}(\boldsymbol{x}'_{i_j}))))$
9:         **end for**
10:         Compute helper example: $\tilde{\boldsymbol{x}}_{i_j} \leftarrow \boldsymbol{x}_{i_j} + 2\,(\boldsymbol{x}'_{i_j} - \boldsymbol{x}_{i_j})$
11:         Set helper label: $\tilde{y}_{i_j} \leftarrow \arg\max_k f_{\boldsymbol{\theta}_{\text{std}}}(\boldsymbol{x}'_{i_j})_k$
12:     **end for**
13:     $\boldsymbol{\theta}_{\text{HAT}} \leftarrow \boldsymbol{\theta}_{\text{HAT}} - \frac{\eta}{m} \cdot \sum_{j=1}^m \nabla_{\boldsymbol{\theta}_{\text{HAT}}} \Big( \text{CE}\,(y_i, f_{\boldsymbol{\theta}_{\text{HAT}}}(\boldsymbol{x}_{i_j})) + \beta \cdot \text{KL}(f_{\boldsymbol{\theta}_{\text{HAT}}}(\boldsymbol{x}_{i_j}), f_{\boldsymbol{\theta}_{\text{HAT}}}(\boldsymbol{x}'_{i_j}))$
$+ \gamma \cdot \text{CE}(\tilde{y}_{i_j}, f_{\boldsymbol{\theta}_{\text{HAT}}}(\tilde{\boldsymbol{x}}_{i_j})) \Big)$
14: **until** training completed

---

contrast to AT, this allows preventing the undesirable excessive rise in the margin to some extent, thus making it possible to achieve significantly better performance on clean samples. Put differently, HAT can also be framed as performing geometric self-distillation (Hinton et al., 2015) to mimic certain geometric properties of a standard trained network.

**HAT algorithm.** We choose to instantiate HAT by extending TRADES which alike TRADES allows us to balance the accuracy-robustness trade-off. Algorithm 1 summarizes the pseudo-code for HAT. The training objective for HAT comprises of three terms: standard loss, robust loss, and an additional helper loss. Thus, in comparison to TRADES, we have an additional parameter $\gamma$ which controls the weight of helper loss and thus, the extent of resistance to excessive directional margin. Finally, note that our extension can be easily plugged into other recent techniques e.g. Wu et al. (2020); Gowal et al. (2021); Rebuffi et al. (2021). In fact, in our experiments, we indeed combine HAT with the approach in Rebuffi et al. (2021) to advance the prior art by a significant margin (Sec. 5.3).

## 5 EXPERIMENTS

This section constitutes an extensive evaluation of HAT. Initially, to test the universality of our approach, we study the performance of HAT with ResNets on different datasets and attack configurations. Next, we leverage extra data and wider networks to obtain state-of-the-art performance on conventional robustness benchmarks. Towards the end, we conduct experiments to analyze HAT.

### 5.1 PERFORMANCE EVALUATION WITH RESNETS

We here empirically validate the performance of HAT. We report results using ResNet-18 (He et al., 2016) on three datasets: CIFAR-10, CIFAR-100 (Krizhevsky, 2009) and SVHN (Netzer et al., 2011). We compare HAT with three prominent adversarial defenses: (i) AT (Madry et al., 2018), (ii) TRADES (Zhang et al., 2019) and (iii) MART (Wang et al., 2020).

**Training setup.** We borrow the set of training hyperparameters from DAWNBench (Coleman et al., 2017). Precisely, we use SGD optimizer with Nesterov momentum (Nesterov, 1983); cyclic learning rates (Smith & Topin, 2018) with cosine annealing and a maximum learning rate of 0.21 for CIFAR-10, CIFAR-100, and 0.05 for SVHN. We train each model for 50 epochs on CIFAR-10 and CIFAR-100 whereas we apply 15 epochs on SVHN. For $\ell_\infty$ training, we use PGD attack with maximum perturbation $\varepsilon = 8/255$ and run the attack for $K = 10$ iterations for all datasets. The PGD step size is set to $\alpha = \varepsilon/4 = 2/255$ for CIFAR-10, CIFAR-100; $\alpha = 1/255$ for SVHN. For HAT, we fix $\gamma$ to

0.5 and use $\beta = 2.5$ for CIFAR-10 and SVHN; $\beta = 3.5$ for CIFAR-100. Whereas the regularization parameter $\beta$ for TRADES is set to 5.0 for CIFAR-10, SVHN and 6.0 for CIFAR-100. For MART, we choose $\beta = 5.0$. More details can be found in App. C.1.

**Evaluating robustness.** During training, we perform early stopping (Rice et al., 2020) and select the model that has the highest robustness against PGD ($K = 20$) for further evaluation. Further, to evaluate the robust accuracy of our models, we use AutoAttack (Croce & Hein, 2020) which comprises an ensemble of four diverse attacks (including a black-box attack). AutoAttack has been consistently shown to provide a reliable estimation of robustness. Nevertheless, we also conduct additional evaluations and checks with HAT to eliminate the possibility of gradient obfuscation in App. C.4.

Table 2: Comparison of HAT using ResNet-18 on CIFAR-10, CIFAR-100 and SVHN with other adversarial defenses under $\ell_\infty$ perturbations of size $\varepsilon = 8/255$. We report the average scores over 3 runs and relegate the standard deviations to Table 6 in App C.3.

| Method | CIFAR-10 | | CIFAR-100 | | SVHN | |
|---|---|---|---|---|---|---|
| | Clean | Robust | Clean | Robust | Clean | Robust |
| Standard | 94.57 | 0.0 | 76.00 | 0.0 | 96.14 | 0.14 |
| AT | 84.01 | 47.74 | 57.50 | 23.88 | 92.57 | 46.33 |
| TRADES | 82.73 | 48.80 | 56.70 | 23.63 | 91.01 | 52.99 |
| MART | 79.52 | 47.98 | 50.82 | 24.52 | 91.30 | 48.46 |
| HAT | 84.90 | 49.08 | 59.19 | 23.75 | 93.08 | 52.83 |

**Results.** Table 2 reports the performance of HAT and other prominent defenses in the literature. It is evident that HAT can notably increase the clean accuracy of the models with little to no degradation of robustness. In other words, HAT consistently shrinks the gap between accuracy and robustness by around 2% compared to prior adversarial training schemes. For example, in the case of CIFAR-10, HAT boosts clean accuracy by 2.17% whilst achieving similar robustness as TRADES. In addition, we find that the 2.17% gain in clean accuracy on CIFAR-10 equates to a 2.08% jump on common corruptions (Hendrycks & Dietterich, 2019) (see App. C.3) which is highly desirable given the practical relevance of common corruptions. Finally, we note that our models do not show any signs of gradient obfuscation as evident from the analysis in App. C.4.

## 5.2 OTHER THREAT MODELS AND LARGER DATASETS

To verify the generality of our results, we now experiment with different attack configurations and larger benchmarks.

**Threat models.** We investigate two different threat configurations on CIFAR-10: (i) $\ell_\infty$ with norm $12/255$ and (ii) $\ell_2$ with norm $128/255$ and follow the same setup as Sec. 5.1. Here, we compare HAT only with TRADES since TRADES achieves superior results than other techniques on CIFAR-10. As shown in Table 3, HAT surpasses TRADES by a large margin. Specifically, with $\ell_\infty$ perturbations of size $12/255$, HAT clearly surpasses TRADES, effecting 5.95%

Table 3: Performance of TRADES and HAT with ResNet-18 on CIFAR-10 under $\ell_p$-constrained adversaries. We report the average results over 3 runs.

| Norm | $\varepsilon$ | Method | Clean | Robust |
|---|---|---|---|---|
| $\ell_\infty$ | 12/255 | TRADES | 73.35 | 32.77 |
| | | HAT | 79.30 | 33.47 |
| $\ell_2$ | 128/255 | TRADES | 87.41 | 68.99 |
| | | HAT | 88.87 | 69.09 |

improvement in clean accuracy while simultaneously bettering robustness by 0.70%. Moreover, HAT achieves a higher clean accuracy ($\uparrow$ 1.46%) against $\ell_2$ perturbations as well. However, the improvement here is rather limited owing to the fact that when training with smaller $\varepsilon$'s, the phenomenon of excessive directional margin might not be very severe.

**Larger datasets.** The efficacy of HAT indeed holds for large-scale datasets such as TinyImageNet-200 and ImageNet-100 (Deng et al., 2009), where AT is known to cause a huge degradation in accuracy. We observe that HAT improves the accuracy of AT from 47.76% to 52.60% and robustness from 17.92% to 18.14% on TinyImageNet-200 against $\ell_\infty$ distortions of size $8/255$. With

Table 4: Comparison of HAT with other state-of-the-art approaches on CIFAR-10 and CIFAR-100 under $\ell_\infty$ adversary. Following Gowal et al. (2021), we report the result of a single run.

| Dataset | Model | Method | Extra data | Clean | Robust |
|---------|-------|--------|-----------|-------|--------|
| CIFAR-10 | PRN-18 | Rebuffi et al. (2021) | DDPM | 83.53 | 56.66 |
| | | HAT | DDPM | 86.86 | 57.09 |
| | | HAT | 80M TI | 89.02 | 57.67 |
| | WRN-28-10 | Rebuffi et al. (2021) | DDPM | 85.97 | 60.73 |
| | | HAT | DDPM | 88.16 | 60.97 |
| | | AWP (Wu et al., 2020) | 80M TI | 88.25 | 60.04 |
| | | Gowal et al. (2021)[3] | 80M TI | 89.48 | 62.80 |
| | | HAT | 80M TI | 91.30 | 62.50 |
| | WRN-34-10 | HAT | 80M TI | 91.47 | 62.83 |
| CIFAR-100 | PRN-18 | Rebuffi et al. (2021) | DDPM | 56.87 | 28.50 |
| | | HAT | DDPM | 61.50 | 28.88 |
| | WRN-28-10 | Rebuffi et al. (2021) | DDPM | 59.18 | 30.81 |
| | | HAT | DDPM | 62.21 | 31.16 |

ImageNet-100 against $\ell_\infty$ perturbations of size $4/255$, HAT outperforms TRADES by $4.20\%$ on clean accuracy and $1.92\%$ on robustness; surpasses AT by $1.36\%$ on clean accuracy and $0.98\%$ on robustness. More details on the experiment can be found in App. C.3.

### 5.3 EFFECT OF ADDITIONAL DATA AND WIDER NETWORKS

It has been observed that robust generalization benefits remarkably from the use of extra training data (Schmidt et al., 2018) and increased model capacity (Madry et al., 2018). We investigate the impact of these two factors on HAT training. Specifically, we build upon the recent approaches from Gowal et al. (2021) and Rebuffi et al. (2021) which provide a carefully designed experimental suite composed of model weight averaging, SiLU activation function, larger models and additional training data to considerably progress the state-of-the-art performance on multiple robustness benchmarks. While Gowal et al. (2021) uses an additional subset of 500k natural images extracted from 80 Million Tiny Images (80M TI) dataset (Torralba et al., 2008), Rebuffi et al. (2021) leverages 1M synthetic images generated by DDPM (Ho et al., 2020). Following the exact experimental setup from these papers[2], we train several models on CIFAR-10 and CIFAR-100 using three different model architectures namely, PreAct ResNet-18 (PRN-18) (He et al., 2016), WideResNet-28-10 (WRN-28-10) and WideResNet-34-10 (WRN-34-10) (Zagoruyko & Komodakis, 2016). However, since the additional data from Gowal et al. (2021) is not publicly available[3], we resort to using a similar dataset provided by Carmon et al. (2019). Note that we always pick $\beta = 3.5$ and $\gamma = 0.5$ for HAT.

In Table 4, we benchmark the performance of HAT along with that of state-of-the-art approaches. Using PRN-18 and synthetic DDPM-generated images, HAT improves clean and robust accuracy by (i) $3.33\%$ and $0.43\%$ respectively on CIFAR-10 and (ii) $4.63\%$ and $0.38\%$ respectively on CIFAR-100 against $\ell_\infty$ perturbations with norm $8/255$. Notably, HAT sets a new state-of-the-art on CIFAR-10 obtaining $89.02\%$ clean accuracy and $57.67\%$ robust accuracy. More importantly, these sizeable improvements with PRN-18 suggest that smaller networks have not yet exhausted their limits and we must invent techniques which better exploit the capacity of these networks. Moreover, with WRN-28-10, HAT also surpasses the state-of-the-art accuracy in the respective categories by 2-3%.

### 5.4 ANALYSIS

**Accuracy vs. robustness trade-off.** We investigate the sensitivity of HAT to its hyperparameters namely, the weight of robust loss $\beta$ and the weight of helper loss $\gamma$. Fig. 5 includes a plot which

---

[2]Note that we do not use CutMix augmentation (Yun et al., 2019) with the setup from Rebuffi et al. (2021).

[3]Gowal et al. (2021) use a custom regenerated dataset which contributes to around $0.7\%$ improvement in robustness in comparison to the data from Carmon et al. (2019). See Sec. 4.3 from Gowal et al. (2021).

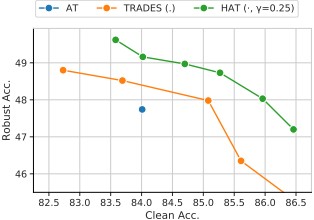

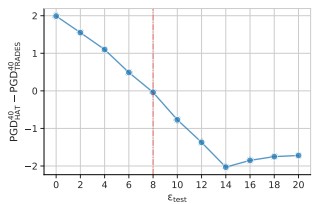

Table 5: Median margin for different methods along $\mathcal{R}_{\text{init}}$. For a fair comparison, the hyperparameters of AT, TRADES and HAT are suitably chosen so that all the three methods have the same robustness of about $47.9\%$ to AutoAttack.

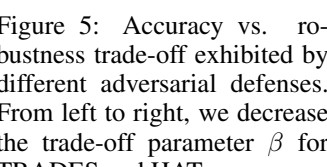

Figure 5: Accuracy vs. robustness trade-off exhibited by different adversarial defenses. From left to right, we decrease the trade-off parameter $\beta$ for TRADES and HAT.

Figure 6: Difference between robust accuracy (PGD$^{40}$) of HAT and TRADES vs. $\varepsilon_{\text{test}}$ ($\ell_\infty$-norm of PGD). The red line corresponds to the value of $\varepsilon$ the models are trained with.

| Algorithm | Median Margin |
|---|---|
| AT | 9.3 |
| TRADES | 9.7 |
| HAT | 9.1 |

demonstrates the performance of HAT at different values of $\beta \in [1.5, 4.0]$ and $\gamma$ fixed to $0.25$. We also show the curve obtained by varying the regularization parameter $\beta$ of TRADES in Fig. 5. We do not include MART here since it achieves a relatively poor trade-off. We can clearly see that for HAT, $\beta$ controls the trade-off between accuracy and robustness where increasing $\beta$ improves robustness. Next, we study the impact of $\gamma$. First, as expected, HAT with $\gamma = 0$ performs identically to TRADES. Second, $\gamma \in \{0.25, 0.5\}$ achieves the best trade-off and within this range, the choice of $\gamma$ has negligible influence on the resulting trade-off exhibited by HAT. Third, increasing $\gamma$ beyond $0.75$ causes the robust accuracy to deteriorate due to the dominance of helper loss over robust loss. The hyperparameter sweeps for $\gamma$ are presented in App. C.5.

**How does HAT improve clean accuracy.** The improvement in the performance on clean samples provided by HAT can be accredited to the following two observations on CIFAR-10: (i) In contrast to TRADES, HAT marginally compromises robustness to $\ell_\infty$ perturbations with a larger norm. We take a robust network and evaluate it using PGD attack ($K = 40$) for different values of $\ell_\infty$-norm $\varepsilon \in [0, 20/255]$. Fig. 6 plots the difference between robustness of HAT and TRADES vs. $\varepsilon$. We observe that while HAT outperforms TRADES at smaller $\varepsilon$'s, it performs slightly worse after the $\varepsilon$ exceeds the value used during training, i.e., $\varepsilon > 8/255$. This implies that we have traded robustness to high $\varepsilon$'s for an improvement in accuracy. (ii) HAT exhibits a slightly lower directional margin along initial adversarial directions $\mathcal{R}_{\text{init}}$ in comparison to AT and TRADES. The margin along $\mathcal{R}_{\text{init}}$ for different algorithms (as evaluated on a random subset of $1024$ samples from the CIFAR-10 test set) is illustrated in Table 5. Note that in Table 5, we list the models that achieve the same robust accuracy ($\sim 47.9\%$ to AutoAttack). Yet, the median margin for HAT is slightly lower than that for AT and TRADES. Additionally, in comparison to TRADES, HAT reduces the number of samples with margin greater than $10\varepsilon$ from $40\%$ to $27\%$.

## 6 CONCLUSION

We presented experimental evidence to highlight that state-of-the-art adversarial defenses foster a superfluous increase in the margin along certain adversarial directions of the input space. This largely destroys the discriminative characteristics along these directions and partly contributes to the much-debated accuracy vs. robustness trade-off. Further, inspired by our analysis, we introduced a novel algorithm, *Helper-based Adversarial Training (HAT)*, to alleviate the problem of excessive directional margin. HAT attempts to mimic the discriminative features learnt by standard trained networks to improve the accuracy on clean samples, hence achieving a superior accuracy vs. robustness trade-off compared to existing defenses. Moreover, HAT surpasses the known state-of-the-art clean accuracy by $3\text{-}5\%$ without sacrificing robustness. Finally, we verify that HAT slightly reduces the directional margin, thus directly benefiting the accuracy. We believe that our experiments and the proposed HAT algorithm can open the door for further research on relieving the accuracy vs. robustness trade-off under limited model capacity. While HAT is only one possible candidate, future works can come up with more effective techniques to alleviate the problem of excessive margin.

ACKNOWLEDGMENTS

The authors would like to thank Alhussein Fawzi and Guillermo Ortiz-Jimenez for their feedback on an earlier version of this paper.

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

## A  TOY PROBLEM

### A.1  EXPERIMENTAL SETUP

Fig. 7 illustrates the toy dataset used for our experiment from two different viewpoints. To be precise, we draw the 3 features $x_1, x_2$ and $x_3$ as follows. $x_1 = \rho_i \cos(z) + \epsilon_1$, $x_2 = \rho_i \sin(z) + \epsilon_2$ and $x_3 \sim \mathcal{U}(\alpha_i, \beta_i)$ where $z \sim \mathcal{U}(0, 2\pi)$ and $\epsilon_1, \epsilon_2 \sim \mathcal{N}(0, \sigma^2)$ where $i = 1, 2$ for class 1 and 2 respectively. We set $\sigma = 0.2$, $\rho_1 = 0.35$, $\rho_2 = 1$, $\alpha_1 = 0.65$, $\beta_1 = 0.70$, $\alpha_2 = 0.80$ and $\beta_2 = 0.85$.

We use a single hidden layer MLP with 25 hidden units and ReLU activation. For training, we use SGD with momentum 0.9, weight decay 0.0005 and set the learning rate to 0.1. We train the model via standard training and adversarial training respectively for 100 epochs. We use $\ell_\infty$ PGD with step size $\alpha = 0.05$, maximum perturbation radius $\varepsilon = 0.1$ and run $K = 5$ iterations for AT.

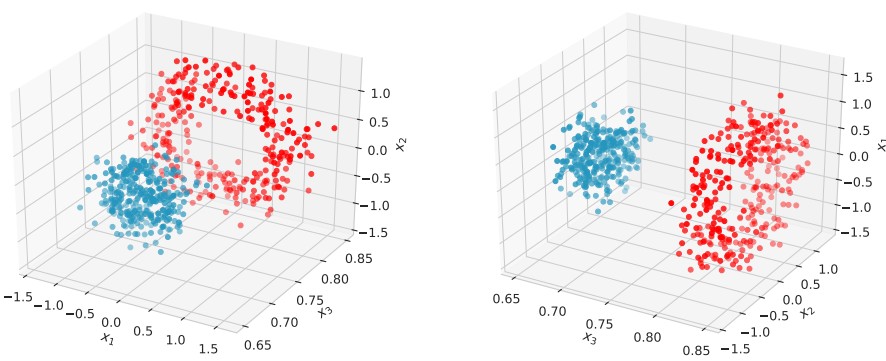

Figure 7: Toy dataset used in our experiment.

### A.2  HAT ALLEVIATES EXCESSIVE MARGIN

In this part, we check whether HAT can correct the failure mode of adversarial training for the toy problem studied in Sec. 3. Fig. 8 shows the classification boundary learnt with HAT. Clearly, the decision boundary in the $x_1$-$x_3$ plane is tilted and not completely parallel to the $x_3$ axis, indicating some dependence on the feature $x_3$. Thus, HAT indeed attenuates the problem of excessive margin which is also reflected in a 4% improvement in accuracy over adversarial training. Additionally, it improves robustness from 73.8% to 74.6%.

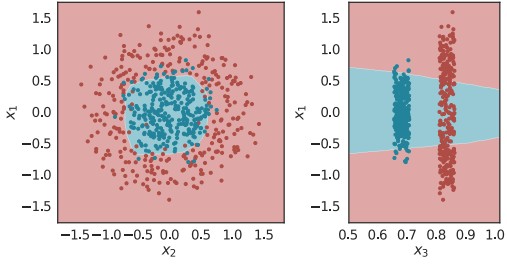

Figure 8: Decision boundary learnt by MLP visualized in two dimensions: $x_1$-$x_2$ and $x_1$-$x_3$ respectively. HAT reduces the margin along $x_3$ and improves accuracy by 4% over AT. For the subplot on the left, we fix $x_3 = 0.85$ and for the right subplot, $x_2 = 0.4$.

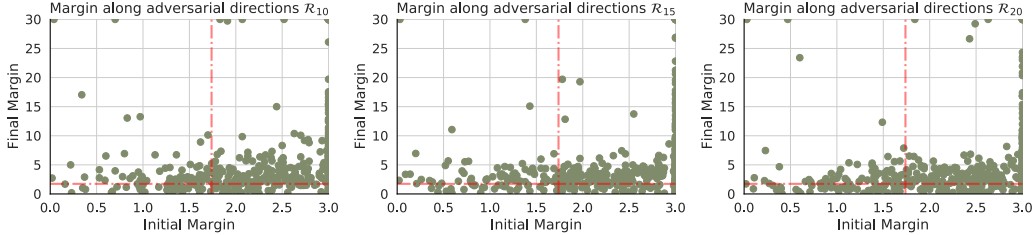

Figure 9: Final margins for adversarially trained model vs. initial margins before the start of adversarial fine-tuning on CIFAR-10 along $\mathcal{R}_{10}$, $\mathcal{R}_{15}$ and $\mathcal{R}_{20}$ respectively. The red dashed line indicates the value of $\varepsilon$ used during training and evaluation.

## B  SEC. 3 CONTINUED: EVIDENCE ON CIFAR-10

### B.1  EXPERIMENTAL SETUP

For the margin experiment, we train ResNet-18 (He et al., 2016) on CIFAR-10 (Krizhevsky, 2009) training set. We simply adopt the set of hyperparameters and some improvements from DAWN-Bench (Coleman et al., 2017) submissions. We use SGD optimizer with Nesterov momentum 0.9 (Nesterov, 1983) and weight decay 0.0005. We further use cyclic learning rates (Smith & Topin, 2018) with cosine annealing and a maximum learning rate of 0.21. We train the model for 50 epochs via standard training. Then, we perform adversarial fine-tuning for 25 epochs with the same scheduler and learning rate settings. Further, we evaluate margins on a set of 512 samples drawn uniformly at random from CIFAR10 test set for the visualizations in Fig. 4 (and Fig. 9).

### B.2  ADDITIONAL MARGIN PLOTS

The margins along other adversarial directions $\mathcal{R}_{10}, \mathcal{R}_{15}, \mathcal{R}_{20}$ before and after adversarial fine-tuning are displayed in Fig. 9. Here, $\mathcal{R}_k$ corresponds to the adversarial directions computed at the $k^{\text{th}}$ iteration of adversarial fine-tuning. We see an increase in the margin along other adversarial directions as expected. Nevertheless, the relative increase along these directions is not as large as compared to that along initial ones $\mathcal{R}_0$ (see Fig. 4).

## C  FURTHER PERFORMANCE EVALUATION

### C.1  EXPERIMENTAL SETUP FOR SEC. 5.1

In this section, we list all the details of our training and evaluation setup. We run all our experiments thrice and report the average scores obtained unless stated otherwise.

**Training setup.** We use ResNet-18 (He et al., 2016) for CIFAR-10 and CIFAR-100 (Krizhevsky, 2009); and PreAct ResNet-18 (He et al., 2016) for SVHN (Netzer et al., 2011). For all our experiments, we use SGD optimizer with Nesterov momentum (Nesterov, 1983), momentum factor 0.9 and weight decay 0.0005. We further employ cyclic learning rates (Smith & Topin, 2018) with cosine annealing and a maximum learning rate of 0.21 for CIFAR-10 and CIFAR-100; 0.05 for SVHN. For CIFAR-10 and CIFAR-100, we train the models for 50 epochs with a batch size of 128. In the case of SVHN, we only train for 15 epochs.

For computing adversarial examples during training, we apply $\ell_\infty$-PGD with the following hyperparameters: $\ell_\infty$ norm $\varepsilon = 8/255$, step size $\alpha = 2/255$ and run the attack for $K = 10$ iterations. Note that we re-implement AT, TRADES and MART and train existing methods and HAT according to the aforementioned settings. The hyperparameters of TRADES, MART and HAT are as follows. For HAT, we fix $\gamma$ to 0.5 and use $\beta = 2.5$ for CIFAR-10 and SVHN; $\beta = 3.5$ for CIFAR-100. The regularization parameter $\beta$ for TRADES is set to 5.0 for CIFAR-10, SVHN and 6.0 for CIFAR-100. For MART, we choose $\beta = 5.0$. We also use the same setup to train a standard model for computing helper labels during HAT training.

Table 6: Comparison of HAT using ResNet-18 on CIFAR-10, CIFAR-100 and SVHN with other adversarial defenses under $\ell_\infty$ adversary ($\varepsilon = 8/255$). We report the average scores over 3 runs.

| Method | CIFAR-10 | | CIFAR-100 | | SVHN | |
|---|---|---|---|---|---|---|
| | Clean | Robust | Clean | Robust | Clean | Robust |
| Standard | $94.57 \pm 0.07$ | $0.0 \pm 0.0$ | $76.00 \pm 0.24$ | $0.0 \pm 0.0$ | $96.14 \pm 0.04$ | $0.14 \pm 0.03$ |
| AT | $84.01 \pm 0.11$ | $47.74 \pm 0.16$ | $57.50 \pm 0.14$ | $23.88 \pm 0.07$ | $92.57 \pm 0.16$ | $46.33 \pm 0.24$ |
| TRADES | $82.73 \pm 0.36$ | $48.80 \pm 0.08$ | $56.68 \pm 0.20$ | $23.63 \pm 0.07$ | $91.01 \pm 0.20$ | $52.99 \pm 0.03$ |
| MART | $79.52 \pm 0.65$ | $47.98 \pm 0.08$ | $50.82 \pm 0.02$ | $24.52 \pm 0.13$ | $91.30 \pm 0.10$ | $48.46 \pm 0.22$ |
| HAT | $84.90 \pm 0.10$ | $49.08 \pm 0.01$ | $59.19 \pm 0.07$ | $23.75 \pm 0.14$ | $93.08 \pm 0.03$ | $52.83 \pm 0.04$ |

**Evaluation protocol.** During training, we perform early stopping (Rice et al., 2020) i.e., we track the robustness of the model to PGD ($K = 20$) on the test set and pick the model that performs the best for further evaluation. In order to benchmark the $\ell_\infty$ robustness, we always test our models against AutoAttack (AA) (Croce & Hein, 2020) using the default code available at https://github.com/fra31/auto-attack.

**Setup for other attack configurations.** We reuse the exact aforementioned setup when investigating different threat models on CIFAR-10. In the case of $\ell_2$ perturbations, we use the following training adversary: $\varepsilon = 128/255$, $\alpha = 15/255$ and $K = 10$. We choose $\beta = 2.5$ and $\gamma = 0.5$ for HAT and $\beta = 5.0$ for TRADES. Whereas we set $\alpha = 4/255, K = 10$ for $\ell_\infty$ perturbations with norm $\varepsilon = 12/255$. Additionally, we pick $\beta = 3.5$ and $\gamma = 0.5$ for HAT and $\beta = 6.0$ for TRADES.

## C.2 EXPERIMENTAL SETUP FOR SEC. 5.3

For training the models enumerated in Table 4 of Sec. 5.3, we follow the setting designed by Gowal et al. (2021) and Rebuffi et al. (2021) respectively. For the sake of completeness, we briefly recap the training setup here. We use SiLU activation function (Hendrycks & Gimpel, 2016) with PreAct ResNet (He et al., 2016)/WideResNet (Zagoruyko & Komodakis, 2016) backbone. The optimizer used is SGD with Nesterov momentum (Nesterov, 1983), momentum factor $0.9$ and weight decay $0.0005$. We further use cyclic learning rates (Smith & Topin, 2018) with cosine annealing, a maximum learning rate of $0.4$ and a warmup of $10$ epochs. The training batch size is set to $1024$ with $70\%$ of the batch comprising of extra/synthetic data. We also use model weight averaging (Izmailov et al., 2018) with decay $\tau = 0.995$ and train the models for $400$ CIFAR-10-equivalent epochs with the extra data from Carmon et al. (2019). In the setting with synthetic DDPM-generated data, we apply $800$ epochs. The training attack used is PGD ($K = 10$) with step size $\alpha = 2/255$ and norm $\varepsilon = 8/255$. Finally, we perform early-stopping by tracking the performance on a disjoint validation set using PGD (K=40) with margin loss (Carlini & Wagner, 2017). We separate first $1024$ samples from the training set for validation. Note that we do not employ CutMix (Yun et al., 2019).

## C.3 ADDITIONAL RESULTS FOR SEC. 5.1

**Detailed version of Table 2.** The complete performance evaluation with ResNet-18 across different datasets appears in Table 6. Here, we also include the standard deviation over 3 runs for completeness. It is clear that HAT outperforms other existing methods, hence narrowing the gap between accuracy and robustness.

**Evaluation with other adversaries.** We additionally scrutinize the performance of robust models trained on CIFAR-10 against conventional adversaries. In order to benchmark the $\ell_\infty$ robustness, we apply PGD with $\varepsilon = 8/255$, $\alpha = 0.01$, $K = 40$ and $r = 5$ restarts following the evaluation protocol used in Carmon et al. (2019), we denote this adversary as PGD+. We also evaluate the vulnerability of our models to PGD with CW loss (Carlini & Wagner, 2017), $\varepsilon = 8/255$, $\alpha = 0.01$ and $K = 40$ steps. Besides, we also test the vulnerability against naturally occurring benign corruptions. Specifically, we use the CIFAR-10-C dataset from Hendrycks & Dietterich (2019). The results presented in Table 7 show that HAT does not break down against other adversaries. Crucially, HAT leads to $2.08\%$ improvement against common corruptions. This suggests that one needs to improve both accuracy as well as robustness in order to advance the performance under real-world distortions.

Table 7: Performance of HAT using ResNet-18 on CIFAR-10 against other $\ell_\infty$ adversaries and common corruptions. We report the average scores over 3 runs.

| Method | Clean | AutoAttack | PGD+ | CW | CIFAR-10-C |
|---|---|---|---|---|---|
| Standard | 94.57 | 0.0 | 0.0 | 0.0 | 72.92 |
| AT | 84.01 | 47.74 | 50.81 | 50.28 | 75.53 |
| TRADES | 82.73 | 48.80 | 52.03 | 49.94 | 74.66 |
| MART | 79.52 | 47.98 | 54.20 | 49.66 | 71.81 |
| HAT | 84.90 | 49.08 | 52.02 | 50.29 | 76.74 |

**Combination with other adversarial defenses.** To study the utility of HAT in concert with recent adversarial defenses, we examine the performance of HAT when integrated with two schemes that improve over TRADES namely, (i) FAT (Zhang et al., 2020) which utilizes early-stopping when constructing adversarial data during training and (ii) AWP (Wu et al., 2020) which encourages flatness in weight loss landscape. Note that although HAT is based on TRADES, it can be easily integrated into these defenses. In our experiments, we follow the same training and evaluation setup as employed by Zhang et al. (2020) and Wu et al. (2020) respectively. The resulting performance on CIFAR-10 is summarized in Table 8. As shown in Table 8, incorporating HAT pushes the clean accuracy of FAT by 2% and also benefits robust accuracy by 0.9%. In combination with AWP, HAT leads to a 2.6% gain in clean accuracy relative to AWP. These results underline that newer methods also suffer from the phenomenon of excessive margin and HAT indeed provides a general fix to alleviate this problem.

Table 8: Performance of HAT (using ResNet-18 on CIFAR-10 against $\ell_\infty$ perturbations of size $8/255$) when combined with two recent adversarial training schemes: FAT (Zhang et al., 2020) and AWP (Wu et al., 2020). We report the scores of a single run.

| Method | Clean | PGD$^{20}$ |
|---|---|---|
| FAT + TRADES | 83.7 | 49.5 |
| HAT + FAT | 85.7 | 50.4 |
| AWP | 82.0 | 55.4 |
| HAT + AWP | 84.6 | 55.4 |

Table 9: Comparison of HAT using PreAct ResNet-18 on TinyImagenet-200 and ImageNet-100 with other adversarial defenses under $\ell_\infty$ adversary. We report the scores of a single run.

| Method | TinyImageNet-200 | | ImageNet-100 | |
|---|---|---|---|---|
| | Clean | Robust | Clean | Robust |
| Standard | 65.02 | 0.0 | 86.71 | 0.0 |
| AT | 47.76 | 17.92 | 75.90 | 49.58 |
| TRADES | 48.25 | 17.17 | 73.06 | 48.64 |
| HAT | 52.60 | 18.14 | 77.26 | 50.56 |

**Larger datasets.** We also examine the performance of HAT on large-scale datasets such as TinyImageNet-200 and ImageNet-100 (Deng et al., 2009). ImageNet-100 (Laidlaw et al., 2021) is a 100-class subset of ImageNet. We use a similar setup as Sec. 5.1 with PreAct ResNet-18 except the following changes. We train for 30 epochs on TinyImageNet-200 against $\ell_\infty$ perturbations of size $8/255$. Whereas on ImageNet-100, we train for 50 epochs with batch size 256, weight decay 0.0001, use input normalization and $\ell_\infty$ perturbations of size $4/255$. On these datasets, we observed that TRADES performed significantly worse than AT. This might be due to the use of KL-divergence loss instead of cross-entropy (CE) loss for crafting adversarial examples. So, we use AT-based formulation of HAT which utilizes CE loss as robust loss as well as for computing adversarial perturbations. For TRADES, we use $\beta = 8.0$ for TinyImageNet-200; $\beta = 6.0$ for ImageNet-100 while we pick $\beta = 1.75$ and $\gamma = 1.0$ for HAT on both the datasets. The results are presented in Table 9. Note that we measure robust accuracy using AutoAttack.

## C.4 CHECKING FOR GRADIENT OBFUSCATION

We evaluate our models with AutoAttack (Croce & Hein, 2020) which has been consistently shown to provide a reliable evaluation of robustness. Nevertheless, we also include additional sanity checks

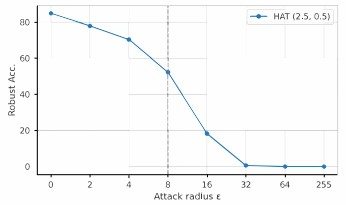

Figure 10: Robust accuracy vs. attack radius $\varepsilon$ of PGD$^{40}$.

Table 10: Robust accuracy vs. number of steps $K$ of PGD$^K$ on CIFAR-10.

| No. of steps $K$ | Robust |
|---|---|
| 40 | 52.34 |
| 200 | 52.30 |
| 500 | 52.24 |
| 1000 | 52.20 |

Table 11: Robust accuracy vs. number of random restarts $r$ of PGD$^{40}$ on CIFAR-10.

| No. of restarts $r$ | Robust |
|---|---|
| 1 | 52.34 |
| 5 | 52.02 |
| 10 | 51.90 |
| 20 | 51.85 |

Table 12: Robust accuracy against transfer black-box PGD$^{10}$ attack on CIFAR-10. For black-box PGD attack, the source model is used to compute adversarial perturbations for evaluating the target model.

| Target model | Source model | | |
|---|---|---|---|
| | HAT | AT | TRADES |
| HAT | - | 62.89 | 61.80 |
| AT | 62.01 | - | 62.20 |
| TRADES | 60.10 | 61.64 | - |

Table 13: Robust accuracy on CIFAR-10 against non-transfer black-box square attack (Andriushchenko et al., 2020) with 5000 queries.

| Model | Square attack (5000 queries) |
|---|---|
| HAT | 56.26 |
| AT | 56.04 |
| TRADES | 55.13 |

to eliminate the possibility of gradient masking. First, following the guidelines in Athalye et al. (2018), we examine the impact of following changes on the robustness of a model trained via HAT on CIFAR-10 against $\ell_\infty$ perturbations of size $8/255$:

- Attack radius $\varepsilon$: Fig. 10 plots the robustness of HAT vs. the radius $\varepsilon$ of PGD ($K = 40$) attack. As expected, increasing the attack budget causes the robust accuracy to monotonically drop to $0\%$. In particular, unbounded PGD adversary reduces the robustness to $0\%$.

- Attack iterations $K$ and random restarts $r$: Increasing the number of attack iterations $K$ or number of random restarts $r$ only marginally lowers the robust accuracy (see Table 10 and Table 11). In order words, the attack has converged and does not suffer from gradient obfuscation.

- As shown in Tables 12 and 13, transfer and non-transfer black box evaluations do not show any signs of gradient obfuscation. We use square attack (Andriushchenko et al., 2020) for non-transfer black box evaluation. Importantly, black box attacks have a lower success rate than white box attacks.

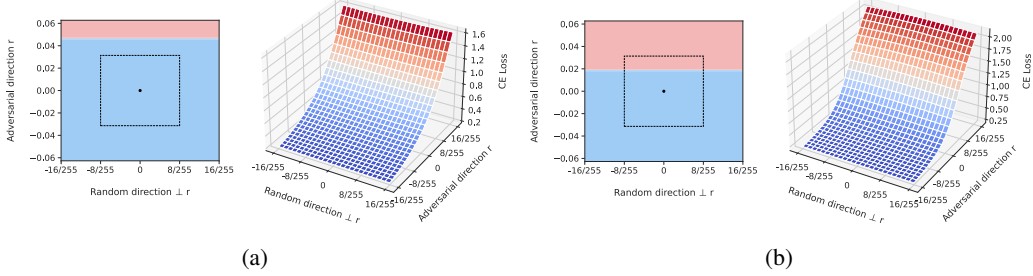

(a)                                                  (b)

Figure 11: Loss landscapes surrounding the 1$^{st}$ and 5$^{th}$ example respectively from CIFAR-10 test set for ResNet-18 trained with $\ell_\infty$ perturbations of size $8/255$. Adversarial direction is the worst-case direction found using PGD$^{20}$ attack. The loss landscapes are very smooth and do not exhibit the typical patterns of gradient obfuscation.

Table 14: Comparison of HAT with other state-of-the-art approaches on CIFAR-10 and CIFAR-100 under $\ell_\infty$ AutoAttack (Croce & Hein, 2020) and GAMA-PGD (Sriramanan et al., 2020) attack.

| Dataset | Model | Method | Extra data | Clean | AutoAttack | GAMA |
|---------|-------|--------|-----------|-------|-----------|------|
| CIFAR-10 | PRN-18 | Rebuffi et al. (2021) | DDPM | 83.53 | 56.66 | 56.81 |
| | | HAT | DDPM | 86.86 | 57.09 | 57.23 |
| CIFAR-100 | PRN-18 | Rebuffi et al. (2021) | DDPM | 56.87 | 28.50 | 28.63 |
| | | HAT | DDPM | 61.50 | 28.88 | 28.99 |

Moreover, as a sanity check, we also visualize the loss landscapes of our trained models in Fig. 11. The loss landscapes are smooth and exhibit low curvature which are the typical characteristics of a robust network (Moosavi-Dezfooli et al., 2019). In summary, these observations indicate that HAT does not lead to gradient obfuscation.

**Evaluation with GAMA-PGD.** To further confirm that our method does not suffer from gradient masking, we investigate the robustness of models reported in Table 4 against GAMA-PGD (Sriramanan et al., 2020), a stronger adversary than vanilla PGD attack. The evaluation results are provided in Table 14. As evident from Table 14, the performance of HAT does not break down even against GAMA attack which signifies that HAT does not cause obfuscated gradients.

## C.5 HAT: ANALYSIS (CONTINUED)

**Accuracy vs. robustness trade-off.** Fig. 12 compares the trade-off obtained by HAT with that of AT and MART (Wang et al., 2020). For HAT, we fix $\gamma = 0.25$ and vary $\beta \in [1.5, 4.0]$; for MART, we vary $\beta \in [1.0, 5.0]$. HAT surpasses AT and MART by a large margin. With MART, we observe notable improvements in robustness to weaker adversaries such as PGD (see Table 7), but the gains diminish when evaluated with a stronger adversary such as AutoAttack (Croce & Hein, 2020).

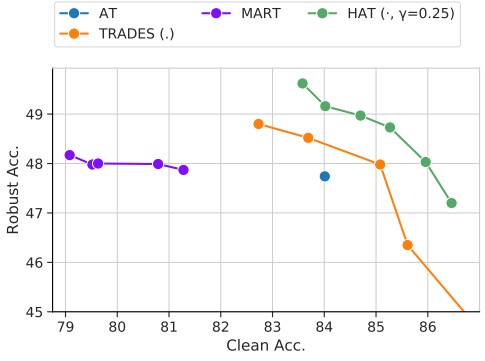

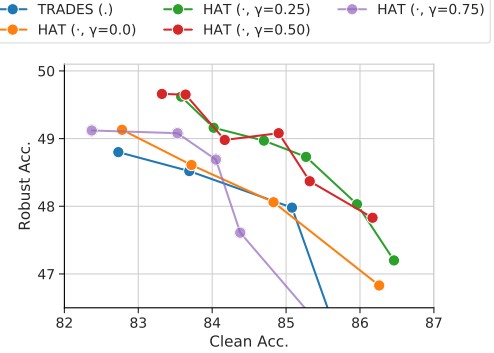

Figure 12: Accuracy vs. robustness trade-off exhibited by AT, TRADES, MART and HAT. From left to right, we decrease the trade-off parameter $\beta$ for TRADES, MART and HAT ($\gamma$ is fixed to 0.25).

Figure 13: HAT accuracy vs. robustness trade-off obtained for different values of $\gamma$. From left to right, we decrease the trade-off parameter $\beta$ for TRADES and HAT. Robust accuracy is evaluated using AutoAttack.

**Sensitivity of HAT to $\gamma$.** We here examine the impact of the weight of helper loss $\gamma$ on the accuracy vs. robustness trade-off exhibited by HAT and conduct additional hyper-parameter sweeps. We show the corresponding trade-off curves for HAT with $\gamma \in \{0.0, 0.25, 0.5, 0.75\}$ in Fig 13. As expected, the trade-off curve for $\gamma = 0$ almost coincides with that of TRADES (note that there is a minor difference due to randomness). Further, $\gamma \in \{0.25, 0.5\}$ achieve the best performance and identical trade-off curves. With $\gamma = 0.75$, the robust accuracy starts to deteriorate due to the dominance of helper loss over robust loss with the effect being more severe for lower values of $\beta$. We believe that increasing $\gamma$ beyond 0.75 would further negatively affect the performance.

Next, we conduct ablation studies on CIFAR-10 ($\ell_\infty$, 8/255) to better understand HAT algorithm. While we elaborate on the HAT design choices below for the sake of completeness, note that we do not conduct any hyperparameter tuning concerning these choices and naively resort to the setting mentioned in Algorithm 1 throughout this work. In the following analysis, we use the same experimental setup as in Sec. 5.1.

**Impact of helper example definition.** We analyze the influence of different choices for defining helper examples for HAT (refer Algorithm 1). We consider the following choices: $x + \alpha r$ where $r$ is the adversarial perturbation computed at $x$ and $\alpha \in \{1.5, 2.0, 2.5, 3.0, 3.5\}$. Note that the helper label is always queried at $x + r$ as in Algorihtm 1. Selecting $\alpha = 2.0$ means that the helper examples are given by $x + 2r$ which represents the setup used in this paper. Fig. 14 illustrates the performance obtained by the resulting HAT-trained models. A lower value for $\alpha$ e.g., $\alpha = 1.5$, hinders the model from increasing the margin and hence, the resulting model achieves high clean accuracy but a slightly poor robust accuracy. $\alpha = 2.0$ and $\alpha = 2.5$ achieve identical results, and the model does not compromise robustness yet achieves a gain in clean accuracy. Further increasing $\alpha$ beyond 3.0 leads to a drop in both clean as well as robust accuracy. This is because the resulting training samples might not be well separated and their target labels may conflict with each other.

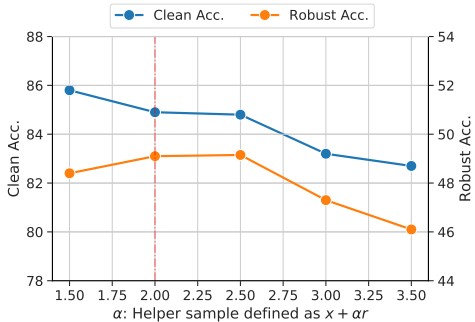

Figure 14: Effect of different helper example definitions. We define a helper example as $x + \alpha r$. Choosing $\alpha = 2.0$ corresponds to the setting used throughout this paper and is indicated by the red dashed line in the plot. Note that the helper label is still queried at $x + r$. We pick $\beta = 2.5$ for HAT.

Figure 15: Effect of different choices for obtaining helper labels. We query helper label $\tilde{y}$, via a standard model $f_{\theta_{std}}$, at $x + \alpha r$. Choosing $\alpha = 1.0$ corresponds to the setting used in this paper and is indicated by the red line. Note that the helper example is still defined as $x + 2r$. (Solid line): We fix $\beta = 2.5$ for HAT. (Dashed): We use a higher $\beta$ with HAT for larger $\alpha$'s.

**Ablation study for obtaining helper labels.** We perform an ablation study with different choices for obtaining the labels for helper examples during HAT training (refer Algorithm 1). Specifically, we query the helper label $\tilde{y}$, via a regularly trained model $f_{\theta_{std}}$, at $x + \alpha r$ where $r$ is the adversarial perturbation computed for input $x$ and vary $\alpha \in \{0.5, 1.0, 1.5, 2.0, 2.5, 3.0, 3.5\}$. In other words, the helper label is defined as the prediction of the standard model $f_{\theta_{std}}$ at $x + \alpha r$; whereas the helper example is always defined to be $x + 2r$. Setting $\alpha = 1.0$ recovers the setup used in this work. The results are shown in Fig. 15 (solid lines). For $\alpha \in \{0.5, 1.0, 1.5\}$, we observe a similar performance relative to that reported in Table 2. With $\alpha \geq 2.0$, the clean accuracy of the robust model slightly increases whilst its robust accuracy starts to degrade. This is because helper labels queried at farther points with $\alpha \geq 2.0$ might be much more pessimistic (worst-case) and hence, present more opposition to the robust loss. Thus, we observe a drop in robust accuracy which can be in fact addressed by simply increasing the weight of robust loss. This is indicated by the dashed lines in Fig. 15 where we increase the weight of robust loss $\beta$ in proportion to the rise in $\alpha$. Taking this adjustment into account leads to the same performance as that with $\alpha = 1.0$, indicating that the underlying trade-off still remains the same.

C.6    COMPARISON WITH CHEN ET AL. (2021)

The work by Chen et al. (2021) is closest our approach in the sense that it also uses knowledge distillation or self-training. However, the standpoint and the way of using it is significantly different from

our work. We use distillation with a motivation to prevent excessive margin and benefit clean accuracy, whilst Chen et al. (2021) uses it from the usual perspective of using self-learning to improve generalization and thereby alleviate robust overfitting. Moreover, it applies self-training at the adversarial inputs while we apply it for overly perturbed helper data. From a computational perspective, the approach from Chen et al. (2021) uses two self-teachers, one of which is adversarially trained, which presents a significant overhead compared to HAT which requires only a regularly trained one. In addition, Chen et al. (2021) use model weight averaging which begs a larger number of training iterations to obtain improvements. Empirically, the distillation setup from Chen et al. (2021) which uses two self-teachers attains $83.67\%$ accuracy and $48.03\%$ robustness against AutoAttack. Clearly, HAT outperforms these results by achieving $84.90\%$ accuracy and $49.08\%$ robustness. Moreover, we also replicate their setup with model weight averaging and perform HAT to further improve performance. In contrast to their scores of $84.65\%$ accuracy and $49.35\%$ robustness, our proposed HAT obtains $85.69\%$ accuracy and $49.34\%$ robustness.

