# OpenReview forum: "Reducing Excessive Margin to Achieve a Better Accuracy vs. Robustness Trade-off"
_ICLR.cc/2022/Conference — ICLR 2022 Poster_

### Official Review · Reviewer_vhey · 2021-10-29

**Correctness:** 3
**Technical Novelty And Significance:** 2
**Empirical Novelty And Significance:** 3
**Recommendation:** 6
**Confidence:** 4

**Main Review:**

Strengths:
- The writting is easy to follow, while the illustration of the idea of HAT is clear and reasonable.
- I especially admire the empirical evaluations in this paper, which involve large-scale experiments using DDPM generated data and 80M TI extra data. The improvements are significant, and the sanity check for, e.g., gradient masking is also presented.

Weaknesses:
- The modifications introduced in HAT are simple (which is good), but they depend on an assumption that ``the model should not be robust beyond the threat model``. Namely, under an 8/255 $\ell_{\infty}$-norm threat model, an adversarial example with 16/255 perturbation is encouraged by HAT to fool the model, while the label of the adversarial example may not change. For me, this assumption is quite ad-hoc, and introducing another standard model $f\_{\theta\_{\textrm{std}}}$ seems not an elegant solution.

In conclusion, I think the pros and cons of this paper are quite clear. Strong empirical evaluations and promising improvements, but the method itself is somewhat ad-hoc and not very principled. So I would like to recommend an acceptance, but the method could be further polished.


**Summary Of The Paper:**

This paper proposes a helper-based adversarial training (HAT) method to alleviate the trade-off between robustness and accuracy. Empirical evaluations are done on several datasets, and under AutoAttack and common corruptions.

**Summary Of The Review:**

Strong empirical evaluations and promising improvements, but the method itself is somewhat ad-hoc and not very principled.

---

> ### Author Response · Authors · 2021-11-19
> **Response to Reviewer vhey**
>
> Dear Reviewer vhey,
>
> Thank you for taking the time to review our work and providing valuable feedback on the manuscript. Below, we address your concerns.
>
>
> **Q. The modifications introduced in HAT are simple (which is good), but they depend on an assumption that the model should not be robust beyond the threat model. Namely, under an 8/255 $\ell_\infty$-norm threat model, an adversarial example with 16/255 perturbation is encouraged by HAT to fool the model, while the label of the adversarial example may not change. For me, this assumption is quite ad-hoc, and introducing another standard model seems not an elegant solution.**
>
> Yes, our method provides an ad hoc solution, yet is backed by results which underline that it works. And importantly, it can be easily integrated with other methods.
>
> The introduction of standard model is supported by our experiments in Sec. 3 which motivate that certain properties of standard model can be transferred to robust models in order to improve their clean accuracy.  Moreover, asking the model to be not robust beyond $2\varepsilon$ brings more benefits than harm. Current robust models trained with $\ell_\infty, 8/255$ already have a poor robustness to large norms $\varepsilon=16/255$, e.g. adversarial training (ResNet-50) with $\ell_\infty, 8/255$ only has about $18$% accuracy at $16/255$ [1]. Slightly compromising this already low robustness at  $16/255$ brings many other advantages such as a good improvement in clean accuracy (Table 2, 3, 4 in Sec. 5) and even better performance on practically-relevant common corruptions (Table 7 in Appendix C. 3). The improvement on common corruptions highlights that the model is indeed robust beyond the threat model in a good sense.
>
> Besides, if one explicitly requires robustness to larger norms such as $\varepsilon=12/255$ or $\varepsilon=16/255$, the best solution would be to use the respective norm for training. Our method also performs well in that regard, and can be used to obtain considerably better results with larger $\varepsilon$ too (as illustrated with $\varepsilon=12/255$ in Table 2).
>
>
> **But the method itself is somewhat ad-hoc and not very principled. So I would like to recommend an acceptance, but the method could be further polished.**
>
> Yes, but the method is supported by our experimental analysis. Apart from the method itself, our goal was to provide insights about an adverse effect caused by adversarial training, namely excessive margin along certain directions that are useful for the classification performance of the network. We further provided an effective solution to this issue which yields a better clean accuracy on multiple benchmarks.  We would be glad if you have any immediate suggestions which could polish the method and the work in general.
>
> ---
>
> [1] https://github.com/MadryLab/robustness

---

### Official Review · Reviewer_dhm9 · 2021-11-01

**Correctness:** 4
**Technical Novelty And Significance:** 4
**Empirical Novelty And Significance:** 4
**Recommendation:** 8
**Confidence:** 5

**Main Review:**

**Strengths**:
* Clean, original and novel idea leading to good experimental results. Very well written paper with a clear story, with clear arguments and experiments to support the story.
* Very extensive experiments in the main paper and in the appendix. It gives a lot of intuitions about the problem and Lp-norm robustness in general.
* The proposed analytic tools are useful beyond the analysis of the proposed algorithm. Big plus for the toy problem giving interesting intuitions, the margin analysis in Figure 4 and the per epsilon analysis in Figure 6.
* The code is attached in the supplementary materials and anyway, the experimental details and code are very well described in the paper. Hence, the paper seems reproducible.

**Weaknesses/Suggestions/Questions**:
1) In the bullet points in page 2 and other parts of the paper, please specify when the accuracy is the "clean" or "robust" accuracy. Otherwise, there is an ambiguity.
2) It would be great to see how the proposed method performs compared to TRADES on larger models such as WRN-70-16. Maybe by fine-tuning an already pre-trained model to avoid expensive computations.
3) In Figure 6, maybe specify that the variable $\epsilon$ on the x-axis is used for the test-time robust accuracy and not the training procedure.
4) In Figure 13 in the appendix, why do the curves TRADES and HAT ($\gamma=0$) do not match while they are the same method? Is the difference due to the variance in the results?
5) (Very optional but curious to check) I would be curious to see the performance of an alternative helper: $x' = x + r + r'$ where $r'$ is the adversarial perturbation computed at $x + r$. In this way, helper samples could possibly look more "natural" rather than when using $x + 2r$, thus possibly improving the final results. It would require twice as more computations but would be interesting to check.

**Summary Of The Paper:**

**Few sentences summary**: the paper proposes a new training loss for adversarial training in the Lp-norm setting. Based on the observation that adversarial training increases the classification margin in a disproportionate manner compared to the nominal training setting, the authors introduce additional samples called "helpers" to reduce the classification margin. The helper samples, which are samples translated further away in the worst adversarial direction, can change labels compared to the adversarial samples. Helper samples get assigned labels from a standardly trained model, thus acting as a constraint coming from this standardly trained network.

Regarding the **results** and **contributions**:
* Novel method using helpers to define a new training loss for adversarial training.
* On par or better results in robust accuracy compared to the SOTA training loss TRADES on CIFAR-10/100, SVHN, TinyImageNet and a subset of ImageNet.
* Much improved results in clean accuracy compared to TRADES, thus reducing the gap between clean and robust accuracy which is primordial for the application of Lp norm models to practical uses.
* Clear analytical tools based on the margin analysis to investigate the proposed method and how/why it works.

**Summary Of The Review:**

Paper enjoyable to read with extensive experiments supporting a clear and novel idea leading to improved results. The authors also propose great analytical tools to investigate their hypothesis. Hence, I vouch for acceptance.

---

> ### Author Response · Authors · 2021-11-19
> **Response to Reviewer dhm9**
>
> Dear Reviewer dhm9,
>
> We appreciate your positive feedback and useful suggestions for our work. We are glad that you enjoyed reading the paper.
>
> We will incorporate your suggestions 1 & 3 in the revised version of the manuscript. We would also like to elaborate on the following:
>
> ---
>
> Q. It would be great to see how the proposed method performs compared to TRADES on larger models such as WRN-70-16. Maybe by fine-tuning an already pre-trained model to avoid expensive computations.
>
> Thanks for the suggestion. We are also pressing for this. But, for now it seems infeasible with our current setup which is largely limited by memory. While the WRN-70-16 models from prior work [1, 2] use large batch sizes of 1024, the maximum batch size we can use is 64, which might produce suboptimal results. We will keep this thread updated if we figure out a way to train a WRN-70-16 model.
>
>
> Q. In Figure 13 in the appendix, why do the curves TRADES and HAT ($\gamma=0$) do not match while they are the same method? Is the difference due to the variance in the results?
>
> Yes, they are similar but do not match exactly because of variance in the results (the HAT ($\gamma=0$) curve is computed only with a single run and averaging over multiple runs should produce more similar curves).
>
>
> Q. I would be curious to see the performance of an alternative helper: x + r + r1 where r1 is the adversarial perturbation computed at x+r. In this way, helper samples could possibly look more "natural" rather than when using x+2r, thus possibly improving the final results. It would require twice as more computations but would be interesting to check.
>
> We did a quick check with your suggested setting. We do not observe any significant difference in the resulting performance compared to the setting used in the paper.
>
> ---
>
> [1] Sven Gowal, Chongli Qin, Jonathan Uesato, Timothy Mann, and Pushmeet Kohli. Uncovering the limits of adversarial training against norm-bounded adversarial examples. arXiv preprint arXiv:2010.03593, 2021.
>
> [2] Sylvestre-Alvise Rebuffi, Sven Gowal, Dan A. Calian, Florian Stimberg, Olivia Wiles, and Timothy Mann. Fixing data augmentation to improve adversarial robustness. arXiv preprint arXiv:2103.01946, 2021.

---

> > ### Comment · Reviewer_dhm9 · 2021-11-19
> > **Response to the authors**
> >
> > I thank the authors for answering my questions. After acknowledging the other reviews, I will maintain my recommendation of '8'. I do not agree with the concerns about the method being ad-hoc or being based on intuition. Not all the methods can be supported mathematically. The paper extensive experiments supporting the proposed method and providing a better intuition about the general problem of adversarial robustness replace quite well a potential mathematical derivation which might bring little to no insight if not done properly.

---

### Official Review · Reviewer_fdwg · 2021-11-02

**Correctness:** 3
**Technical Novelty And Significance:** 3
**Empirical Novelty And Significance:** 4
**Recommendation:** 6
**Confidence:** 5

**Main Review:**

Strengths:
* The paper is well written and easy to understand. The motivation behind the design choices is clear. All the related works are properly addressed and the baselines are also strong.
* The paper achieves a significant boost as compared to existing methods on strong attacks like Auto-Attack. The approach shows consistent gains across multiple datasets.

Weaknesses:

* I think the results shown in Figure-4 are quite expected as initially when the perturbations are generated from a standard trained model, they will be non-smooth similar to random noise. Thus the final model will have high invariance to the directions of these random noise as compared to the perturbations which are smooth in nature and have features. These smoother perturbations would be generated by the adversarially trained models and thus the model would be easily fooled as we go in the direction of these perturbations. This is addressed by the Final Margin of figure 4-c. Although the proposed approach reduces the invariance in the directions of the initial perturbations which are similar to random noise, (as shown in table 1, 5 and stated in section 3), I think ideally the model should focus on reducing the invariance in the direction of smooth perturbations which have semantic features. Could the authors clarify a bit on this. I dont think it would matter much if the model will reduce the invariance in the directions of initial perturbations(similar to random noise) since they won't change the semantics of the image to some other class image. While the smooth perturbations which have some semantics and are generated using an adversarial model have the potential to change the semantics of an image and thus change the true class of the image as well as shown in [1] and thus it is desired to reduce invariance in these directions.

Some minor concerns:

* Could the authors clarify how they plotted the class boundaries in figure 3? I think this is plotted by examining the predictions of all the points possible in the 3D space?
* In table 5 it is shown that the models are trained so that they have the same robustness. I think this is not a good idea for an ideal comparision. Could the authors show the same table with the median margin in R-init , R-5 and R-15 where the models do not have any constraint on having the same robust accuracy. If possible could the authors share the results of table-1 for R-5 and R-15 also.
* I think the training budget for the results reported in Table 6 is only 50 epochs. If possible could the authors share the results of HAT for all three datasets for 200 epochs training budget? This will help in better understanding the proposed approach. I think the activation used without additional data is ReLU. If this is true could the authors also share the CIFAR10 200 epochs without additional data results for SiLU activation. In case the authors have used SiLU can they share the results with RelU.
* If possible could the authors share the PRN18 and WRN-28-10 for CIFAR10 and CIFAR100(if possible) results as shown in table 4 using the ReLU activation? This would help in understanding the influence of SiLU activation.
* An ablation study on using different perturbation bounds for getting the helper label in Algorithm-1 can also help a lot in better understanding the proposed approach.

[1] Tramèr, F., Behrmann, J., Carlini, N., Papernot, N., & Jacobsen, J. (2020). Fundamental Tradeoffs between Invariance and Sensitivity to Adversarial Perturbations. ArXiv, abs/2002.04599.

**Summary Of The Paper:**

The paper highlights the presence of excessive invariance in the prediction of robust models along the initial adversarial directions. Initial adversarial directions refers to the directions in which adversarial images generated using a standard trained model are present. Based on this hypothesis the authors propose a training method where the excessive invariance is minimized using the cross entropy loss between the prediction(made using the standard trained model) of larger epsilon adversarial image and the prediction of the adversarial image, in addition to the TRADES loss formulation. This additional loss term indeed improves the accuracy-robustness trade-off by giving a significant boost in clean accuracy along with a slight boost in adversarial robustness as compared to the existing methods. Overall the paper is well written and easy to follow.

**Summary Of The Review:**

Overall I think the paper is well written. It shows a significant boost as compared to existing art and has some minor issues at present.
If the concerns are properly addressed I am willing to increase my score.

---

> ### Author Response · Authors · 2021-11-19
> **Response to Reviewer fdwg**
>
> Dear Reviewer fdwg,
>
> Thank you for providing valuable feedback on the manuscript. We are glad that you found the paper well written and easy to follow. We answer your concerns below:
>
>
> **Q. Could the authors clarify how they plotted the class boundaries in figure 3? I think this is plotted by examining the predictions of all the points possible in the 3D space?**
>
> Yes, to plot the decision boundaries, we evaluated the predictions of all possible points in 3D space.
>
>
> **Q. I think the results shown in Figure-4 are quite expected as initially when the perturbations are generated from a standard trained model, they will be non-smooth similar to random noise. Thus the final model will have high invariance to the directions of these random noise as compared to the perturbations which are smooth in nature and have features. These smoother perturbations would be generated by the adversarially trained models and thus the model would be easily fooled as we go in the direction of these perturbations. This is addressed by the Final Margin of figure 4-c. Although the proposed approach reduces the invariance in the directions of the initial perturbations which are similar to random noise, (as shown in table 1, 5 and stated in section 3), I think ideally the model should focus on reducing the invariance in the direction of smooth perturbations which have semantic features. Could the authors clarify a bit on this. I do not think it would matter much if the model will reduce the invariance in the directions of initial perturbations(similar to random noise) since they won't change the semantics of the image to some other class image. While the smooth perturbations which have some semantics and are generated using an adversarial model have the potential to change the semantics of an image and thus change the true class of the image as well as shown in [1] and thus it is desired to reduce invariance in these directions.**
>
> First, the perturbations generated from a standard model, which we call initial adversarial directions, although may look unintuitive to humans; they are certainly not random directions. In fact, as supported by our analysis, they contain strongly discriminative features which are essential for the classification performance of the network. This is very much in-line with recent literature [2][3] which outlines that these directions contain useful features. Exhibiting a large invariance along those directions negatively affects the clean accuracy of the network. Hence, reducing invariance along these definitely matters since it can perhaps improve clean accuracy without compromising robustness. This is in fact corroborated by our results. Second, you are right that the later adversarial perturbations are smoother and introduce semantic changes which look meaningful to humans. It might also be beneficial to reduce invariance along these, but, the invariance here is already not very severe (shown in Fig. 4c). Hence, in our paper, we focus on the invariance along initial adversarial directions which are closely tied to the good classification performance of standard training.
>
>
> **Q. In table 5 it is shown that the models are trained so that they have the same robustness. I think this is not a good idea for an ideal comparision. Could the authors show the same table with the median margin in R-init , R-5 and R-15 where the models do not have any constraint on having the same robust accuracy. If possible could the authors share the results of table-1 for R-5 and R-15 also.**
>
> Different methods like AT & TRADES have slightly different robust accuracy (cf. Table 2: 47.7% for AT vs. 48.8% for TRADES). Since the level of robust accuracy also has some influence on the margin (see Table 1), it might not be very fair to directly compare these models. A more unbiased comparison would be to make sure the models under consideration have the same robust accuracy (which can be easily ensured by choosing an appropriate $\beta$ for TRADES and HAT) so that we can compare the margin in isolation. The idea is, for a given level of robustness, the margin should be as small as possible (but larger than $\varepsilon$) to get improvements on clean accuracy. So, we believe that the comparison in Table 5 suits this goal. Let us know if this addresses your concerns on the fairness of the comparison. **[Continued below]**

---

> > ### Author Response · Authors · 2021-11-22
> > **Response to Reviewer fdwg: Further clarification (Continued 2)**
> >
> > Dear Reviewer fdwg,
> >
> > Below, we provide additional evaluations for the margin analysis, as requested. Let us know if this clarifies your concerns.
> >
> > **[Continued] Q. In table 5, it is shown that the models are trained so that they have the same robustness. I think this is not a good idea for an ideal comparision. Could the authors show the same table with the median margin in R-init , R-5 and R-15 where the models do not have any constraint on having the same robust accuracy. If possible could the authors share the results of table-1 for R-5 and R-15 also.**
> >
> > The margin evaluations along R-init, R-5 and R-15 for the models in Tables 1 & 5 respectively are presented below. Note that there might be a small variance in the results owing to randomness (compared to those in the paper) since we had to re-run some of the experiments because we did not have checkpoints for the required models.
> >
> > First, from Tables 1 & 5 below, it is evident that the margin along R-5 and R-15 (about 2.25x $\varepsilon_\text{train}$) is not so severe as opposed to that along R-init (5.5x $\varepsilon_\text{train}$). Besides, R-init being the adversarial directions for a standard network, are highly essential for its classification performance, and certainly are not random directions. Hence, we say that reducing the invariance along R-init can improve the clean accuracy of robust models, which is indeed corroborated by our results in Sec. 5.
> >
> > [Table 1]: TRADES: We could only manage to re-run some of the models from Table 1. We hope that they still convey the message.
> >
> > | $\beta$ | Rinit | R5 | R15 | Clean | Robust |
> > |:----:|:----:|:----:|:----:|:----:|:----:|
> > | 1.0 | 8.3 | 4.0 | 3.7 | 88.1 | 43.8 |
> > | 3.0 | 9.8 | 4.4 | 4.0 | 84.6 | 47.9 |
> > | 5.0 | 10.5 | 4.5 | 4.0 | 82.7 | 48.8 |
> >
> > Second, as shown in Table 5 below, HAT reduces margin along R-init relative to TRADES, hence achieves 2.2% improvement in clean accuracy. But, compared to AT, its R-init margin is slightly higher; this is because HAT has about 1.4% higher robust accuracy than that of AT, and robust accuracy does also influence R-init margin (higher robust accuracy $\implies$ larger R-init margin cf. Table 1). Therefore, we argued that a more fair ground would be to choose an appropriate $\beta$ for TRADES and HAT so that the resulting models have the same level of robustness and we can independently compare margin. This is the way we compare in Table 5 in the paper where all models have the same robust accuracy of ~47.9%, but HAT has a lower margin along R-init, and thus pushes clean accuracy by about 2% over that of AT & TRADES.
> >
> > [Table 5]: As requested, results with no constraint on having the same robust accuracy. We use the same parameters as those used for the models in Table 2.
> >
> > |Algorithm|R-init|R-5|R-15|Clean|Robust|
> > |:---|:---:|:---:|:---:|:---:|:---:|
> > | AT | 9.3 | 4.4 | 3.9 |  84.1 | 47.7 |
> > | TRADES | 10.5 | 4.5 | 4.0 | 82.7 | 48.8 |
> > | HAT | 9.5 | 4.4 | 4.0 | 84.9 | 49.1 |
> >
> > Also, note the following subtle differences with [1]. We aim to reduce invariance beyond the $\ell_p$-norm ball. Whereas [1] focus on invariance within the $\ell_p$-norm ball which leads to invariance-based examples that lie in the $\ell_p$ ball itself, yet are labelled differently by a human oracle, questioning the choice $\ell_p$-norm ball as imperceptibility constraint. Besides [1] mostly experiment with MNIST, and we believe that their notion of invariance might be less relevant on other datasets such as CIFAR-10 because $\ell_\infty, 8/255$ changes are usually less perceptible on CIFAR-10 (than $\ell_\infty, 0.3$ on MNIST) which makes them less capable to cause such invariance-based examples.
> >
> > ---
> >
> > [1] Tramèr, F., Behrmann, J., Carlini, N., Papernot, N., & Jacobsen, J. (2020). Fundamental Tradeoffs between Invariance and Sensitivity to Adversarial Perturbations. ArXiv, abs/2002.04599.

---

> > > ### Comment · Reviewer_fdwg · 2021-11-25
> > > **Thanks for the response**
> > >
> > > I really appreciate the authors for providing additional results. However, most of the concerns that I raised are not properly clarified. The main contribution of the paper is the improved robustness and clean accuracy when using additional data. However, I think it's a bit strange, why the authors could not get improved robustness when additional data is not there. Further, I think that the approach is not reducing the margin in the directions of perturbations generated by an adversarially trained model. In order to improve the generalization, I feel it is necessary to reduce the margin in these directions in the first place because these perturbations can change the true label of the image and cause conflict in the training objective. I think more study needs to be done regarding this. Therefore I am not fully convinced with the approach. However, provided the empirical novelty I retain my score and am happy to suggest acceptance.

---

> > > > ### Author Response · Authors · 2021-11-25
> > > > **Additional Clarification for Reviewer fdwg**
> > > >
> > > > Thank you for your feedback. We are keen to fully address your questions. We provide a short elaboration below which hopefully resolves your concerns.
> > > >
> > > > **Why the authors could not get improved robustness when additional data is not there.**
> > > >
> > > > We believe that this is a new concern which was not brought up during the rebuttal phase. Nevertheless, we would like to reiterate that *the goal of our method is to improve clean accuracy while being robust*, and we never claim to build a method to substantially improve robustness over other methods. Yet, the robust accuracy of our models is still competitive and even better than that of other methods in most scenarios.
> > > >
> > > > **I feel it is necessary to reduce the margin along adversarial directions for an adversarially trained model in the first place.**
> > > >
> > > > With due respect, our experimental analysis contradicts your hypothesis. Our analysis in Sec. 3 suggests that reducing the excessive margin along “adversarial directions for a standard network (R-init)” can potentially benefit clean accuracy. Whereas, the margin along “adversarial directions for an adversarially trained model (say R-15)” is already not so severe as indicated by [Table 1] in [our previous response](https://openreview.net/forum?id=Azh9QBQ4tR7&noteId=I-RV0Y81Cmk) . For instance, from [Table 1], the margin along R-init for TRADES ($\beta=5.0$) is about $6\varepsilon_{\text{train}}$, while that along R-15 is only $2.3\varepsilon_{\text{train}}$. Thus, we try to resolve the problem of insensitivity along R-init in the first place which consequently improves clean accuracy. This completes our argument.
> > > >
> > > > The significance of R-init for classification accuracy is also well-supported in the current literature [1][2] and is elegantly summarized by the following excerpt from [2]: “The input image-space directions along which the networks are most vulnerable to attack are the same directions which they use to achieve their classification performance in the first place.” Whereas there is little support to the fact the reducing margin along R-15 can benefit generalization.
> > > >
> > > > Let us know if you have any other concerns which prevent you from increasing your score.
> > > >
> > > > ---
> > > >
> > > > [1] Andrew Ilyas, Shibani Santurkar, Dimitris Tsipras, Logan Engstrom, Brandon Tran, and Aleksander Madry. Adversarial examples are not bugs, they are features. In Advances in Neural Information Processing Systems, 2019
> > > >
> > > > [2] Saumya Jetley, Nicholas Lord, and Philip Torr. With friends like these, who needs adversaries? In Advances in Neural Information Processing Systems, 2018

---

> > > > > ### Author Response · Authors · 2021-11-27
> > > > > **Questions & Clarification for Reviewer fdwg**
> > > > >
> > > > > Dear Reviewer fdwg,
> > > > >
> > > > > Thank you for engaging with us. As we are approaching the end of the discussion period, we would like to kindly remind you to consider our recent response to your concerns and express your views on it.
> > > > >
> > > > > Additionally, we carefully looked at your responses again to ensure that we completely understand your argument. It would be great if you can shortly elaborate on the following points directly taken from your initial response:
> > > > >
> > > > > **1.** You say that **“The final model will have high invariance to the directions of these random noise (R-init) as compared to the perturbations which are smooth in nature and have features. These smoother perturbations (R-15) would be generated by the adversarially trained models and thus the model would be easily fooled as we go in the direction of these perturbations. This is addressed by the Final Margin of Figure 4-c.”**
> > > > >
> > > > > It seems that you already agree with us on the fact that invariance along adversarial directions for an adversarially trained network (R-15) is not severe, while that along R-init is much more severe. Thus, the introduction of helpers deals with this excessive margin along R-init.
> > > > >
> > > > > **2.**  You say that **“I don't think it would matter much if the model will reduce the invariance in the directions of initial perturbations (R-init) (similar to random noise).”**
> > > > >
> > > > > Your hypothesis is contradicted by Sec 3 in our paper and also previously published works [1, 2]. We would like to re-stress that *R-init directions are not random noise* and instead have been shown to matter for the impressive classification performance of a standard network. In fact, a standard network is much more robust to random noise, while has zero robustness to R-init.
> > > > >
> > > > > It would be great if you can clarify/elaborate on these two points as we believe they can address your concerns.
> > > > >
> > > > > Thanks!
> > > > >
> > > > > Authors
> > > > >
> > > > > ---
> > > > > [1] Andrew Ilyas, Shibani Santurkar, Dimitris Tsipras, Logan Engstrom, Brandon Tran, and Aleksander Madry. Adversarial examples are not bugs, they are features. In Advances in Neural Information Processing Systems, 2019
> > > > >
> > > > > [2] Saumya Jetley, Nicholas Lord, and Philip Torr. With friends like these, who needs adversaries? In Advances in Neural Information Processing Systems, 2018

---

> > > > > ### Comment · Reviewer_fdwg · 2021-11-29
> > > > > **Reply to authors**
> > > > >
> > > > > Kindly note that I am aware that the adversarial perturbations generated by an adversarially trained model are not random noise. However if you will try to visualize them, then they will be similar to random noise. Therefore though they have some features but they can't change the true label of an image. On the other side if we look at the perturbations generated by R-15 then they will be much more smooth and they will have more features. They can therefore change the true label of the image. Even a margin of 2.3*epsilon that is around19/255 is also significant and can change the true label. In order to get a better understanding I would suggest the authors to visualize CIFAR-10 images:
> > > > >  1) Generated from a standard trained model for epsilon=16,24,32
> > > > > 2) Generated from an adversarially trained model for epsilon=16,24,32
> > > > >
> > > > > I think the authors will find that for the first case the true label won't change while it will change in the second case(change is more clear for larger epsilon).
> > > > > Thus based on this I think that the model should definitely have a lower margin in R-init but since these cant change the true label, it is more important to reduce the margin in R-15, on which I am not convinced with the results presented by the authors. In the final version, I think the authors should try to analyze this. Finally, I would congratulate the authors for their amazing work!!

---

> > > > > > ### Author Response · Authors · 2021-11-29
> > > > > > **Closing Remarks for Reviewer fdwg**
> > > > > >
> > > > > > Thanks for taking the time to respond to our clarifications. We are delighted to know that you found our work to be amazing. We believe that both of us have converged on most points. Below, we summarize our discussion in this thread and also add more details.
> > > > > >
> > > > > > We think that your definition of features mainly entails human-plausible features; otherwise you would not say R-15 has more features (and are more important) than R-init. Although R-init looks unintuitive (random in your words) to us, technically both R-init and R-15 contain features from the perspective of standard and adversarially trained (AT) networks respectively.
> > > > > >
> > > > > > >In summary, *since R-init contain features from the standpoint of a standard network, they are more significant for clean accuracy. Hence, we deal with the excessive margin along R-init which is well reflected by the theme of this paper.*
> > > > > >
> > > > > > Further, we definitely agree with you that R-15 with large $\varepsilon$’s ($\geq 16/255$) can potentially change the true label for a human. But, at least for now, we (and previous works) do not have any evidence to suggest that R-15 can considerably improve clean accuracy beyond that of AT. We believe that this problem is orthogonal to that along R-init and we never claimed to have dealt with it. Whereas we claim to alleviate the problem with R-init via our approach.
> > > > > >
> > > > > > Once again, we appreciate your timely responses and thank you for engaging with us!
> > > > > >
> > > > > > ---
> > > > > >
> > > > > > Note: For correctness, the adversarial perturbations computed for AT model are R-25 since the model is trained for 25 epochs; R-15 are perturbations found during the intermediate part of training. The final margin along R-25 is even lower - only $1.72\varepsilon_{\text{train}}$ which is large but not excessive.

---

> ### Author Response · Authors · 2021-11-19
> **Response to Reviewer fdwg (Continued)**
>
> **Q. I think the training budget for the results reported in Table 6 is only 50 epochs. If possible could the authors share the results of HAT for all three datasets for 200 epochs training budget? This will help in better understanding the proposed approach. I think the activation used without additional data is ReLU. If this is true could the authors also share the CIFAR10 200 epochs without additional data results for SiLU activation. In case the authors have used SiLU can they share the results with RelU. If possible could the authors share the PRN18 and WRN-28-10 for CIFAR10 and CIFAR100(if possible) results as shown in table 4 using the ReLU activation? This would help in understanding the influence of SiLU activation.**
>
> Yes, the training budget for the results in Table 6 is only 50 epochs. But, note that we use a cyclic learning rate schedule and not the usual waterfall schedule. Note that this schedule is not specific to this paper and has been increasingly used in recent papers [4]. Our cyclic schedule has been tuned to achieve the same performance as that of training with a waterfall schedule with a large number of epochs (100 to 200). Since adversarial training is computationally heavy, using the cyclic schedule allows us to significantly reduce the training time while the resulting models still have the same performance. For reference, on CIFAR-10 $\ell_\infty$, 8/255, our TRADES model achieves 82.7% clean and 48.8% robust accuracy, which is similar to the results obtained by a carefully-tuned setup in [5] with much more epochs: 81.5% clean and 49.1% robust (cf. Table 13, row 3 in [5]). Thus, training for 200 epochs should not make any difference.
>
> The activation used without additional data is ReLU (Tables 2, 3) and with additional data is SiLU (Table 4) following [6]. An insightful analysis of ReLU vs. SiLU activation is already conducted in [6] which shows that SiLU leads to about 1% gain in robust accuray over ReLU with WideResNets. (Figure 5 in [6]).
>
> We hope that this addresses your questions on the training schedule and activation function. If not, let us know and we would be happy to conduct more experiments to clarify this.
>
>
> **Q. An ablation study on using different perturbation bounds for getting the helper label in Algorithm-1 can also help a lot in better understanding the proposed approach.**
>
> Thanks for the helpful suggestion. Upon your request, we conducted an ablation study with different choices for getting the perturbation label, namely: we query the helper label at $x + \alpha r$ where $\alpha \in$ \{$0.5, 1.0, 1.5, 2.0, 2.5, 3.0, 3.5$\} and $\alpha=1.0$ is the setup used in the paper. The results are shown in the [figure here](https://imgur.com/mLxkqBA). For $\alpha \in $\{$0.5, 1.0, 1.5$\}, we get similar performance as that reported in the paper. With $\alpha \geq 2.0$, the clean accuracy of the robust model slightly increases whilst its robust accuracy undergoes degradation. **[Update: 22 Nov]**: This is because helper labels queried at farther points with $\alpha \geq 2.0$ might be much more pessimistic (worst-case) and hence, present more opposition to the robust loss. Thus, we observe a drop in robust accuracy which can be in fact addressed by simply increasing the weight of robust loss. This is indicated by the dashed lines in the [figure](https://imgur.com/mLxkqBA) where we increase the weight of robust loss $\beta$ in proportion to the rise in $\alpha$. Taking this into account leads to the same performance as that with $\alpha=1.0$, indicating that the underlying trade-off still remains the same.
>
> ---
>
> [1] Tramèr, F., Behrmann, J., Carlini, N., Papernot, N., & Jacobsen, J. (2020). Fundamental Tradeoffs between Invariance and Sensitivity to Adversarial Perturbations. ArXiv, abs/2002.04599.
>
> [2] Andrew Ilyas, Shibani Santurkar, Dimitris Tsipras, Logan Engstrom, Brandon Tran, and Aleksander Madry. Adversarial examples are not bugs, they are features. In Advances in Neural Information Processing Systems, 2019
>
> [3] Saumya Jetley, Nicholas Lord, and Philip Torr. With friends like these, who needs adversaries? In Advances in Neural Information Processing Systems, 2018
>
> [4] Maksym Andriushchenko, and Nicolas Flammarion. Understanding and Improving Fast Adversarial Training. In Advances in Neural Information Processing Systems, 2020
>
> [5] Tianyu Pang, Xiao Yang, Yinpeng Dong, Hang Su, and Jun Zhu. Bag of tricks for adversarial
> training. In International Conference on Learning Representations, 2021.
>
> [6] Sven Gowal, Chongli Qin, Jonathan Uesato, Timothy Mann, and Pushmeet Kohli. Uncovering the limits of adversarial training against norm-bounded adversarial examples. arXiv preprint arXiv:2010.03593, 2021.

---

### Official Review · Reviewer_P4Wc · 2021-11-08

**Correctness:** 4
**Technical Novelty And Significance:** 3
**Empirical Novelty And Significance:** 4
**Recommendation:** 6
**Confidence:** 4

**Main Review:**

### Strengths:
- Extremely simple method, which can be useful for practitioners.
- A slight improvement over baseline defences on CIFAR-10 and CIFAR-100 datasets.

### Weaknesses:
- The method is based on intuition and the authors didn't provide any theoretical justifications for the proposed defence. Based on my intuition, I believe the method is fundamentally flawed as its assumptions are incorrect. For example, it is incorrect to assume that all adversarial examples with perturbations $2 \epsilon$ should be labelled with its adversarial label.
- The authors should compare the robustness of their method for moderate size perturbations as well, e.g. $\epsilon = 12/255$ and $\epsilon = 16/255$ on CIFAR-10 and CIFAR-100. It is quite likely that their method will be less robust for moderate size perturbations.
- The overall procedure is ad-hoc and requires training and storing the model trained without any regularization first. The model is then finetuned with the proposed training procedure.
- Some references are missing and the comparison is outdated. The method should also be compared with [1], [2] and [3] defenses, which improve upon Trades defense.
- The experimental comparison can be improved. The authors evaluated the models with AutoAttack. The authors can also compare their method against GAMA [4] attack. The authors should also include the gradient masking checks in the experimental results or at least discuss gradient masking.

[1] Amirreza Shaeiri, Rozhin Nobahari, and Mohammad Hossein Rohban. Towards deep learning models resistant to large perturbations. arXiv preprint arXiv:2003.13370, 2020.

[2] Jingfeng Zhang, Xilie Xu, Bo Han, Gang Niu, Lizhen Cui, Masashi Sugiyama, and Mohan Kankanhalli. Attacks which do not kill training make adversarial learning stronger. In International Conference on Machine Learning, pp. 11278–11287. PMLR, 2020.
[3] Dongxian Wu, Shu-Tao Xia, and Yisen Wang. Adversarial weight perturbation helps robust generalization. Advances in Neural Information Processing Systems (NeurIPS), 2020.

[4] Gaurang Sriramanan, Sravanti Addepalli, Arya Baburaj, and R Venkatesh Babu. Guided Adversarial Attack for Evaluating and Enhancing Adversarial Defenses. In Advances in Neural Information Processing Systems (NeurIPS), 2020.

### Update after the author's response
The authors addressed all my concerns. In particular, the authors:
- Added adversarial robustness results with $\epsilon = 12$.
- Added adversarial robustness results with other attacks.

Overall, based on the new results for the larger perturbations and the author's comments to other reviewers, I am discarding my doubts about the paper's approach, that it is somewhat ad-hoc. I believe the empirical contributions of this work are significant and novel. Therefore, I recommend accepting the revised paper.

**Summary Of The Paper:**

The authors aim to reduce the gap between clean accuracy without adversarial training and with adversarial training. To improve the robustness-accuracy trade-off, the authors introduce Helper-based Adversarial Training. The main idea is to use adversarial examples $\mathbf{x}_{\text{adv}} = \mathbf{x} + 2 \mathbf{r}$, where $\mathbf{r}$ is standard PGD adversarial perturbation, as helper adversarial examples. The model is trained to classify these helper adversarial examples as the adversarial label predicted by the model trained without adversarial training. In the experiments, the authors show that HAT improves clean accuracy and robust accuracy on CIFAR-10 and CIFAR-100 datasets when compared with TRADES defense.

**Summary Of The Review:**

The authors proposed a simple technique to improve clean accuracy. However, the method is based on intuition, which in my opinion, is flawed: not all large perturbations should be labelled with its adversarial label. The authors should provide a theoretical justification for their intuition. Besides that, the experimental comparison is outdated with few recent defenses missing, which improve upon TRADES defense.

### Update after the author's response
I sincerely thank the authors for addressing the majority of my comments and concerns. The experimental results are undeniable and clearly demonstrate the advantages of the proposed technique. Based on the new results for the medium perturbation $\epsilon = 12$ and additional experiments with other attacks, I tend to overlook my doubts about the paper's approach.

I recommend accepting the revised version of the manuscript.

---

> ### Author Response · Authors · 2021-11-19
> **Response to Reviewer P4Wc**
>
> Dear Reviewer P4Wc,
>
> Thank you for acknowledging the simplicity and usefulness of our method and also expressing your concerns. We address your specific questions below.
>
>
> **Q. The method is based on intuition and the authors didn't provide any theoretical justifications for the proposed defense. Based on my intuition, I believe the method is fundamentally flawed as its assumptions are incorrect. For example, it is incorrect to assume that all adversarial examples with perturbations $2\varepsilon$ should be labelled with its adversarial label.**
>
> As rightly pointed out, our method is based on intuition yet is backed by extensive experimental observations and analysis supporting this intuition. We agree that labeling all helper examples $2\varepsilon$ with adversarial labels might not be the best solution, but it still provides an effective strategy that works in practice as shown by our results in Sec. 5. Moreover, our method does not violate the threat model that we assume & our assumptions are not flawed. For our method, we use the following threat model: $\ell_\infty$ perturbations with norm $\varepsilon$ where we usually choose $\varepsilon=8/255$, which has been the common practice in adversarial robustness literature [5][6]. This threat model dictates that one needs to be robust to perturbations bounded by norm $\varepsilon$, and has no say about robustness to perturbations beyond this threat model i.e., beyond $\varepsilon$. Thus, in this sense, it is not flawed to assign different labels to overly perturbed helper samples (of norm $2\varepsilon$) since they are not contained in the threat model and the trained model will, anyways, have poor robustness at  $2\varepsilon$. For instance, even adversarial training (ResNet-50) with $\varepsilon=8/255$ only has a robustness of about $18$% at $2\varepsilon=16/255$ (as in Table 2 in [1]).
>
> Besides, training using a threat model with attack norm $\varepsilon$ is not the optimal way to obtain robustness to perturbations with twice the norm $2\varepsilon$. If we explicitly desire robustness to larger norms such as $\varepsilon=12/255$ or $\varepsilon=16/255$, we should directly use the respective threat model for training as also done in [1]. Our method also performs well in that realm, and can be used to obtain better results with larger $\varepsilon$ too (as illustrated with $\varepsilon=12/255$ in Table 2.
>
>
> **Q. The authors should compare the robustness of their method for moderate size perturbations as well, e.g. and on CIFAR-10 and CIFAR-100. It is quite likely that their method will be less robust for moderate size perturbations.**
>
> We in fact provide a comparison with TRADES on CIFAR-10 (cf. Fig. 6) where we compare the robustness of each method to $\varepsilon \in [0, 20/255]$. Compared to TRADES, HAT, as expected, is slightly less robust to perturbations with higher norm $\varepsilon \in [12/255, 20/255]$. But, this slight compromise brings other advantages such as a good improvement in clean accuracy (Table 2, 3, 4 in Sec. 5) and even better performance on practically-relevant common corruptions (Table 7 in Appendix C.3), which is highly desirable.
>
>
> **Q. The overall procedure is ad-hoc and requires training and storing the model trained without any regularization first.**
>
> Yes, the procedure only requires a regularly trained model which is easy to have on most datasets owing to the availability of pre-trained models.
>
>
> **Q. Some references are missing and the comparison is outdated. The method should also be compared with [1], [2] and [3] defenses, which improve upon TRADES defense.**
>
> We do not believe that our comparison is outdated. The aim of the comparison in Table 1 was to examine how incorporating helper examples performs over primitive baselines such as adversarial training and TRADES, and understand its effect in isolation without involving other sophisticated heuristics.  Thus, the comparison in Table 1 was not meant to be a comparison with state-of-the-art (SOTA). The SOTA comparison is in fact provided in Table 4 where we compare with the community verified & peer-reviewed SOTA methods (note that SOTA works still rely on TRADES) [5][6], showing superiority of HAT. Notably, in Table 4, we also compared with [3].
>
> Besides, we have already cited and acknowledged [2] and [3] in our work (Sec 1.1, Sec 5.3 Table 4). Since we mostly looked at community-verified approaches, we missed [1] and would cite it aptly in Sec 1.1 of our work. Please let us know if any other references are missing. We would be glad to include them in our paper, if relevant to the topic.
>
> Additionally, we would like to remark that HAT can be integrated with the approaches in [1], [2] and [3] to get additional gains Below, we provide an example combining HAT with [2], [3] on CIFAR-10 with ResNet-18:
>
> |Threat|Method|Clean|PGD-20|
> |:--:|:--|:--:|:--:|
> |$\ell_\infty$, 8/255|FAT-TRADES [2] |83.7|49.5|
> |$\ell_\infty$, 8/255|FAT-HAT|**85.7**|**50.4**|

---

> > ### Author Response · Authors · 2021-11-19
> > **Response to Reviewer P4Wc (Continued)**
> >
> > [Continued]
> >
> > |Threat|Method|Clean|PGD-20|
> > |:--:|:--|:--:|:--:|
> > |$\ell_\infty$, 8/255|AWP [3] |82.0|55.4|
> > |$\ell_\infty$, 8/255|AWP-HAT|**84.6**|**55.4**|
> >
> > In combination with [2] and [3], HAT improves the clean accuracy by $2$% and $2.6$% respectively. This indicates that current methods too suffer from excessive margin and incorporating HAT alleviates the problem in general.
> >
> > **Q. The authors should include the gradient masking checks in the experimental results or at least discuss gradient masking. The experimental comparison can be improved. The authors evaluated the models with AutoAttack. The authors can also compare their method against GAMA [4] attack.**
> >
> > We indeed discussed gradient masking when we evaluated our methods in Sec. 5.1 and performed extensive checks to discard the possibility of gradient masking with HAT. The checks can be found in Appendix C. 4. We observed no patterns of gradient obfuscation with HAT in our analysis.
> >
> > In our opinion, AutoAttack is currently the strongest attack and is strictly better than GAMA [4] even amongst the results in their own paper [4] (see Table 2 in [4]). Nonetheless, upon your request, we evaluate some of the models from Table 2 with GAMA. Here too, we observe that AA is better than GAMA and again we do not see any signs of gradient masking with HAT.
> >
> > |Model|Threat|Data|Method|Clean|AutoAttack|GAMA|
> > |:--:|:--:|:--:|:--:|:--:|:--:|:--:|
> > | PreActResNet-18 | $\ell_\infty$, 8/255 | CIFAR-10 | HAT | 83.5 | 57.1 | 57.2 |
> > | PreActResNet-18 | $\ell_\infty$, 8/255 | CIFAR-10 | [6] | 86.9 | 56.7 | 56.8 |
> > | PreActResNet-18 | $\ell_\infty$, 8/255 | CIFAR-100 | HAT | 56.9 | 28.9 | 29.0 |
> > | PreActResNet-18 | $\ell_\infty$, 8/255 | CIFAR-100 | [6] | 61.5 | 28.5 | 28.6 |
> >
> > We would be happy if you have any other suggestions to improve the experimental comparison. Let us know if this addresses your concerns.
> >
> > ---
> >
> > [1] Amirreza Shaeiri, Rozhin Nobahari, and Mohammad Hossein Rohban. Towards deep learning models resistant to large perturbations. arXiv preprint arXiv:2003.13370, 2020.
> >
> > [2] Jingfeng Zhang, Xilie Xu, Bo Han, Gang Niu, Lizhen Cui, Masashi Sugiyama, and Mohan Kankanhalli. Attacks which do not kill training make adversarial learning stronger. In International Conference on Machine Learning, pp. 11278–11287. PMLR, 2020.
> >
> > [3] Dongxian Wu, Shu-Tao Xia, and Yisen Wang. Adversarial weight perturbation helps robust generalization. Advances in Neural Information Processing Systems (NeurIPS), 2020.
> >
> > [4] Gaurang Sriramanan, Sravanti Addepalli, Arya Baburaj, and R Venkatesh Babu. Guided Adversarial Attack for Evaluating and Enhancing Adversarial Defenses. In Advances in Neural Information Processing Systems (NeurIPS), 2020.
> >
> > [5] Sven Gowal, Chongli Qin, Jonathan Uesato, Timothy Mann, and Pushmeet Kohli. Uncovering the limits of adversarial training against norm-bounded adversarial examples. arXiv preprint arXiv:2010.03593, 2021
> >
> > [6] Sylvestre-Alvise Rebuffi, Sven Gowal, Dan A. Calian, Florian Stimberg, Olivia Wiles, and Timothy Mann. Fixing data augmentation to improve adversarial robustness. arXiv preprint arXiv:2103.01946, 2021.

---

### Author Response · Authors · 2021-11-22
**Thanks and Revision Summary**

We thank all the reviewers for their helpful feedback.  We have revised our manuscript incorporating the suggestions of the reviewers.

Following is a summary of changes to the paper:
- Minor fixes based on suggestions from Reviewer dhm9.
- Cited [1] in Sec. 1.1  (Related work) as asked by Reviewer P4Wc.

Below, we summarize the changes to the Appendix:
- Added evaluation with GAMA-PGD in Appendix C. 4, as requested by Reviewer P4Wc.
- Combination of HAT with other adversarial defenses is provided in Appendix C. 3, which highlights the efficacy of HAT as a general fix to alleviate the problem of excessive margin (Reviewer P4Wc).
- Following suggestion from Reviewer fdwg, an ablation study for the choice of helper examples & helper labels in HAT is included in Appendix C. 5.

Other concerns have been individually addressed in the detailed responses to each review.

---

[1] Amirreza Shaeiri, Rozhin Nobahari, and Mohammad Hossein Rohban. Towards deep learning models resistant to large perturbations. arXiv preprint arXiv:2003.13370, 2020.

---

### Decision · Program_Chairs · 2022-01-20

**Decision:**

Accept (Poster)

**Comment:**

The authors propose a simple addition to adversarial training methods that improves model performance without significantly changing the complexity of training.  The initial reviews raised some questions about whether experiments were sufficiently extensive, but these issues were resolved during the rebuttal and discussion period, resulting in a strong consensus that the paper should be published.